# Chromerid genomes reveal the evolutionary path from photosynthetic algae to obligate intracellular parasites

Yong H Woo[1]*, Hifzur Ansari[1], Thomas D Otto[2], Christen M Klinger[3†], Martin Kolisko[4†], Jan Michálek[5,6†], Alka Saxena[1†‡], Dhanasekaran Shanmugam[7†], Annageldi Tayyrov[1†], Alaguraj Veluchamy[8†§], Shahjahan Ali[9¶], Axel Bernal[10], Javier del Campo[4], Jaromír Cihlář[5,6], Pavel Flegontov[5,11], Sebastian G Gornik[12], Eva Hajdušková[5], Aleš Horák[5,6], Jan Janouškovec[4], Nicholas J Katris[12], Fred D Mast[13], Diego Miranda-Saavedra[14,15], Tobias Mourier[16], Raeece Naeem[1], Mridul Nair[1], Aswini K Panigrahi[9], Neil D Rawlings[17], Eriko Padron-Regalado[1], Abhinay Ramaprasad[1], Nadira Samad[12], Aleš Tomčala[5,6], Jon Wilkes[18], Daniel E Neafsey[19], Christian Doerig[20], Chris Bowler[8], Patrick J Keeling[4], David S Roos[10], Joel B Dacks[3], Thomas J Templeton[21,22], Ross F Waller[12,23], Julius Lukeš[5,6,24], Miroslav Oborník[5,6,25], Arnab Pain[1]*

*For correspondence: yong. woo@kaust.edu.sa (YHW); arnab. pain@kaust.edu.sa (AP)

†These authors contributed equally to this work

Present address: ‡Vaccine and Infectious Disease Division, Fred Hutchinson Cancer Research institute, Seattle, United States; §Biological and Environmental Sciences and Engineering Division, Center for Desert Agriculture, King Abdullah University of Science and Technology, Thuwal, Saudi Arabia; ¶The Samuel Roberts Noble Foundation, Ardmore, United States

Competing interests: The authors declare that no competing interests exist.

[1]Pathogen Genomics Laboratory, Biological and Environmental Sciences and Engineering Division, King Abdullah University of Science and Technology, Thuwal, Saudi Arabia; [2]Parasite Genomics, Wellcome Trust Sanger Institute, Wellcome Trust Genome Campus, Cambridge, United Kingdom; [3]Department of Cell Biology, University of Alberta, Edmonton, Canada; [4]Canadian Institute for Advanced Research, Department of Botany, University of British Columbia, Vancouver, Canada; [5]Institute of Parasitology, Biology Centre, Czech Academy of Sciences, České Budějovice, Czech Republic; [6]Faculty of Sciences, University of South Bohemia, České Budějovice, Czech Republic; [7]Biochemical Sciences Division, CSIR National Chemical Laboratory, Pune, India; [8]Ecology and Evolutionary Biology Section, Institut de Biologie de l'Ecole Normale Supérieure, CNRS UMR8197 INSERM U1024, Paris, France; [9]Bioscience Core Laboratory, King Abdullah University of Science and Technology, Thuwal, Saudi Arabia; [10]Department of Biology, University of Pennsylvania, Philadelphia, United States; [11]Life Science Research Centre, Faculty of Science, University of Ostrava, Ostrava, Czech Republic; [12]School of Botany, University of Melbourne, Parkville, Australia; [13]Seattle Biomedical Research Institute, Seattle, United States; [14]Centro de Biología Molecular Severo Ochoa, CSIC/Universidad Autónoma de Madrid, Madrid, Spain; [15]IE Business School, IE University, Madrid, Spain; [16]Centre for GeoGenetics, Natural History Museum of Denmark, University of Copenhagen, Copenhagen, Denmark; [17]European Bioinformatics Institute (EMBL-EBI), Wellcome Genome Campus, Hinxton, Cambridge, United Kingdom; [18]Wellcome Trust Centre For Molecular Parasitology, Institute of Infection, Immunity and Inflammation, College of Medical, Veterinary and Life Sciences, University of Glasgow, Glasgow, United Kingdom; [19]Broad Genome Sequencing and Analysis Program, Broad Institute of MIT and Harvard, Cambridge, United States; [20]Department of Microbiology, Monash University, Clayton, Australia; [21]Department of Microbiology and Immunology, Weill Cornell Medical College, New York, United States; [22]Department of Protozoology, Institute of Tropical Medicine, Nagasaki University, Nagasaki, Japan; [23]Department of Biochemistry, University of Cambridge, Cambridge, United Kingdom; [24]Canadian Institute for Advanced Research, Toronto, Canada; [25]Institute of Microbiology, Czech Academy of Sciences, České Budějovice, Czech Republic

**Abstract** The eukaryotic phylum *Apicomplexa* encompasses thousands of obligate intracellular parasites of humans and animals with immense socio-economic and health impacts. We sequenced nuclear genomes of *Chromera velia* and *Vitrella brassicaformis*, free-living non-parasitic photosynthetic algae closely related to apicomplexans. Proteins from key metabolic pathways and from the endomembrane trafficking systems associated with a free-living lifestyle have been progressively and non-randomly lost during adaptation to parasitism. The free-living ancestor contained a broad repertoire of genes many of which were repurposed for parasitic processes, such as extracellular proteins, components of a motility apparatus, and DNA- and RNA-binding protein families. Based on transcriptome analyses across 36 environmental conditions, *Chromera* orthologs of apicomplexan invasion-related motility genes were co-regulated with genes encoding the flagellar apparatus, supporting the functional contribution of flagella to the evolution of invasion machinery. This study provides insights into how obligate parasites with diverse life strategies arose from a once free-living phototrophic marine alga.

**eLife digest** Single-celled parasites cause many severe diseases in humans and animals. The apicomplexans form probably the most successful group of these parasites and include the parasites that cause malaria. Apicomplexans infect a broad range of hosts, including humans, reptiles, birds, and insects, and often have complicated life cycles. For example, the malaria-causing parasites spread by moving from humans to female mosquitoes and then back to humans.

Despite significant differences amongst apicomplexans, these single-celled parasites also share a number of features that are not seen in other living species. How and when these features arose remains unclear. It is known from previous work that apicomplexans are closely related to single-celled algae. But unlike apicomplexans, which depend on a host animal to survive, these algae live freely in their environment, often in close association with corals.

Woo et al. have now sequenced the genomes of two photosynthetic algae that are thought to be close living relatives of the apicomplexans. These genomes were then compared to each other and to the genomes of other algae and apicomplexans. These comparisons reconfirmed that the two algae that were studied were close relatives of the apicomplexans.

Further analyses suggested that thousands of genes were lost as an ancient free-living algae evolved into the apicomplexan ancestor, and further losses occurred as these early parasites evolved into modern species. The lost genes were typically those that are important for free-living organisms, but are either a hindrance to, or not needed in, a parasitic lifestyle. Some of the ancestor's genes, especially those that coded for the building blocks of flagella (structures which free-living algae use to move around), were repurposed in ways that helped the apicomplexans to invade their hosts. Understanding this repurposing process in greater detail will help to identify key molecules in these deadly parasites that could be targeted by drug treatments. It will also offer answers to one of the most fascinating questions in evolutionary biology: how parasites have evolved from free-living organisms.

## Introduction

The phylum *Apicomplexa* is comprised of eukaryotic, unicellular, obligate intracellular parasites, infecting a diverse range of hosts from marine invertebrates, amphibians, reptiles, birds to mammals including humans. More than 5000 species have been described to date, and over 1 million apicomplexan species are estimated to exist (*Adl et al., 2007*; *Pawlowski et al., 2012*). Clinically and economically important apicomplexan pathogens, for example, *Babesia*, *Cryptosporidium*, *Eimeria*, *Neospora*, *Theileria*, *Toxoplasma* (*Tenter et al., 2000*), and the malaria-causing parasite *Plasmodium* wreak profound negative impacts on animal and human welfare.

Despite their diverse host tropism (*Roos, 2005*) and life cycle strategies, apicomplexans possess several unifying molecular and cellular features, including the abundance of specific classes of nucleic acid-binding

proteins with regulatory functions in parasitic processes (*Campbell et al., 2010*; *Flueck et al., 2010*; *Radke et al., 2013*; *Kafsack et al., 2014*; *Sinha et al., 2014*), extracellular proteins for interactions with the host (*Templeton et al., 2004a*; *Anantharaman et al., 2007*), an apical complex comprising a system of cytoskeletal elements and secretory organelles (*Hu et al., 2006*), an inner membrane complex (IMC) derived from the alveoli (*Eisen et al., 2006*; *Kono et al., 2012*; *Shoguchi et al., 2013*), and a non-photosynthetic secondary plastid, termed the apicoplast (*McFadden et al., 1996*). How and when these features arose is unclear, owing to the lack of suitable outgroup species for comparative analyses.

Chromerids comprise single-celled photosynthetic colpodellids closely associated (and likely symbiotic) with corals (*Cumbo et al., 2013*; *Janouškovec et al., 2013*). Phylogenetic analysis demonstrates that these algae are closely related to *Apicomplexa* (*Janouškovec et al., 2013*), confirming the long-standing hypothesis that apicomplexan parasites originated from a free-living, photosynthetic alga (*McFadden et al., 1996*; *Moore et al., 2008*). Two known chromerid species, *Chromera velia* and *Vitrella brassicaformis* (*Moore et al., 2008*; *Oborník et al., 2011*, *2012*), can be cultivated in the laboratory, and their plastid (*Janouškovec et al., 2010*) and mitochondrial genomes (*Flegontov et al., 2015*) have been described. We explored whole nuclear genomes of *Chromera* and *Vitrella* to understand how obligate intracellular parasitism has evolved in *Apicomplexa*.

## Results and discussion

### Genome assembly and annotation

A shotgun approach was used to sequence and assemble the *Chromera* and *Vitrella* nuclear genome into 5953 and 1064 scaffolds totaling 193.6 and 72.7 million base-pairs (Mb). The disparity in genome size is attributable largely to the presence of transposable elements (TEs) totaling ~30 Mb in *Chromera* vs only 1.5 Mb in *Vitrella*, as the predicted number of protein-coding genes is almost the same at 26,112 and 22,817, respectively. Detailed characterizations of the two genomes and their gene structures are described in Appendix 1 and *Supplementary files 1, 2*.

### Ancestral gene content of free-living and parasitic species

We constructed a phylogenetic tree of 26 species, comprising *Chromera*, *Vitrella*, 15 apicomplexans, 2 dinoflagellates, 2 ciliates, 4 stramenopiles, and a green alga. On the phylogenetic tree (*Figure 1A*), *Chromera* and *Vitrella* formed a group closest to the apicomplexan clade, consistent with previous phylogenies (*Moore et al., 2008*; *Janouškovec et al., 2010*, *2013*, *2015*; *Oborník et al., 2012*). The long branches from their common node are consistent with drastic differences in morphology, life cycle (*Oborník et al., 2012*), plastid (*Janouškovec et al., 2010*) and mitochondrial genomes (*Flegontov et al., 2015*) between the two chromerids (*Figure 1A*). Likewise, despite common origins, apicomplexans show extensively diverse lifestyles, including host tropism and invasion phenotypes (*Figure 1B*).

We reconstructed the parsimonious gene repertoires for the ancestors of the 26 species, at the nodes of the phylogenetic tree (*Figure 2A*; *Figure 2—figure supplement 1*). We note five key nodes on the evolutionary paths to present-day apicomplexans: the alveolate ancestor; the common ancestor of *Apicomplexa* and chromerids, termed the proto-apicomplexan ancestor; the apicomplexan ancestor; the ancestor of apicomplexan lineages, for example, coccidia and hematozoa; and extant apicomplexans (*Figure 2A*). Protein-coding genes from the 26 species were clustered by OrthoMCL (*Li et al., 2003*) into groups of homologous genes, hereafter defined as orthogroups. We note that an orthogroup could have homologous genes from different species (putative orthologs) or from the same species (putative paralogs arising from gene duplications). Gains or losses of orthogroups are displayed as green or red sections of a pie on the phylogenetic tree in *Figure 2A*. Divergence of the proto-apicomplexan ancestor from the alveolate ancestor (Stage I) was accompanied by losses of 1668 and gains of 2197 orthogroups (sum of the two 'pies' in Stage I). Transition of the free-living proto-apicomplexan ancestor to the apicomplexan ancestor (Stage II) is accompanied by many gene losses (3862 orthogroups) but few gains (81 orthogroups) (*Figure 2A*). Divergence of coccidians, for example, *Toxoplasma gondii*, from the apicomplexan ancestor (Stage III) is characterized by modest changes (537 losses; 414 gains), whereas divergence of hematozoans, for example, *Plasmodium* spp., is marked by drastic losses (1384 losses; 77 gains) (*Figure 2A*). Further divergence of apicomplexan taxa beyond Stage III is characterized by modest, lineage-specific gains (*Figure 2A*). Functional composition of gained genes at various stages will be discussed in later sections. Paucity of gained genes (81 orthogroups) during Stage II indicates that the genome of the

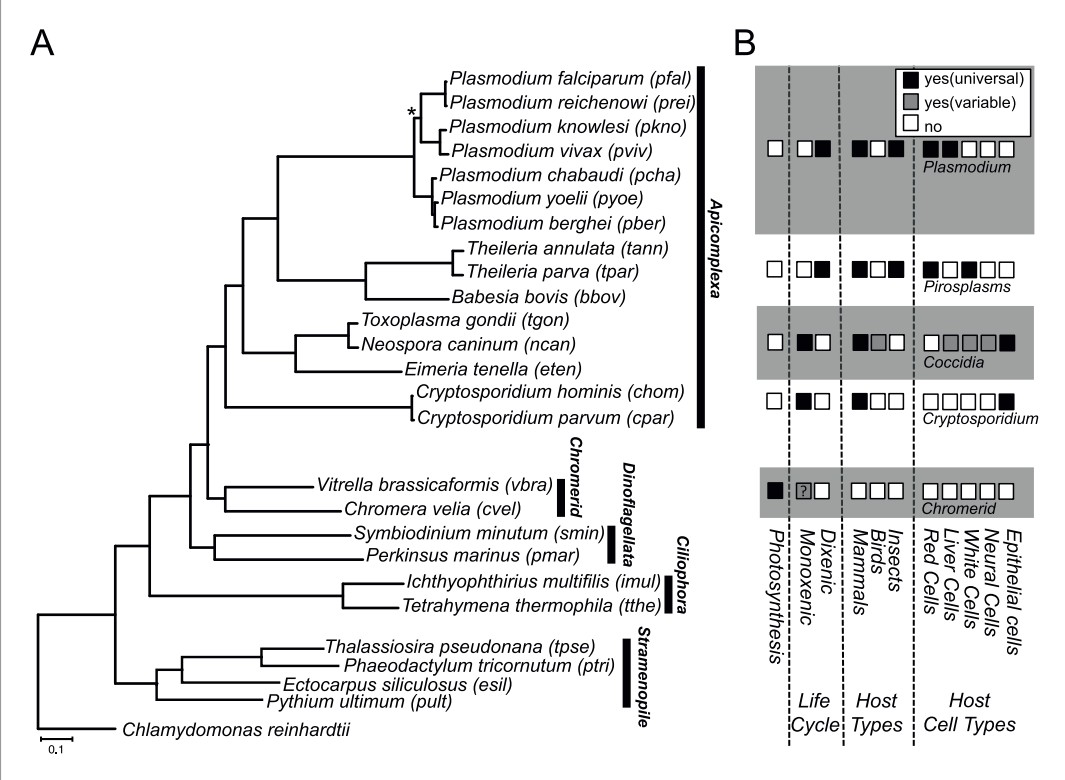

**Figure 1**. Phylogenetic, parasitological, and genomic context of chromerids. (**A**) Phylogenetic tree of 26 alveolate and outgroup species (see *Figure 1—source data 1* for the list of species). Multiple sequence alignments of 101 genes, which have 1:1 orthologs across all species (*Figure 1—source data 2*) were concatenated to a single matrix of 33,997 aligned amino acids. A maximum likelihood tree was inferred using RAxML with 1000 bootstraps, with *Chlamydomonas reinhardtii* as an outgroup. All clades are supported with bootstrap values of 100% except one node (*) with 99%, and also with 1.00 posterior probability from a bayesian phylogenetic tree based on PhyloBayes (*Lartillot and Philippe, 2004*) (CAT-GTR). (**B**) Lifestyles of the apicomplexan and chromerid species under investigation. '?': uncertainty due to lack of relevant data.

The following source data are available for figure 1:

**Source data 1**. List of 24 species excluding Chomera and Vitrella used in this study and their data sources.
**Source data 2**. A list of 101 shared orthogroups with a single gene in all of the 26 species, used for the species phylogenetic tree.

free-living ancestor possessed most of the genes that were present in the common ancestor of apicomplexans and survived in their present-day descendants.

## Progressive, lineage-specific losses during apicomplexan evolution

Parasite evolution has been associated with genome reduction across several branches of the tree of life (*Keeling, 2004*; *Sakharkar et al., 2004*; *Morrison et al., 2007*). Examples also exist, however, where parasite genomes are not reduced (*Pombert et al., 2014*) but expanded (*Raffaele and Kamoun, 2012*), underscoring the fact that the genome reduction process during parasite evolution is not completely understood. We sought to characterize in detail the dynamics of gene loss across apicomplexan evolution, particularly for components of molecular processes that are hallmarks of free-living lifestyle. We performed a systematic analysis of the cellular components involved in: (1) cellular metabolic pathways; (2) the endomembrane trafficking systems, regulating the movement of molecules across intracellular compartments in eukaryotes (*Leung et al., 2008*); and (3) the flagellum, a highly conserved apparatus for motility in aqueous environment (*Silflow and Lefebvre, 2001*).

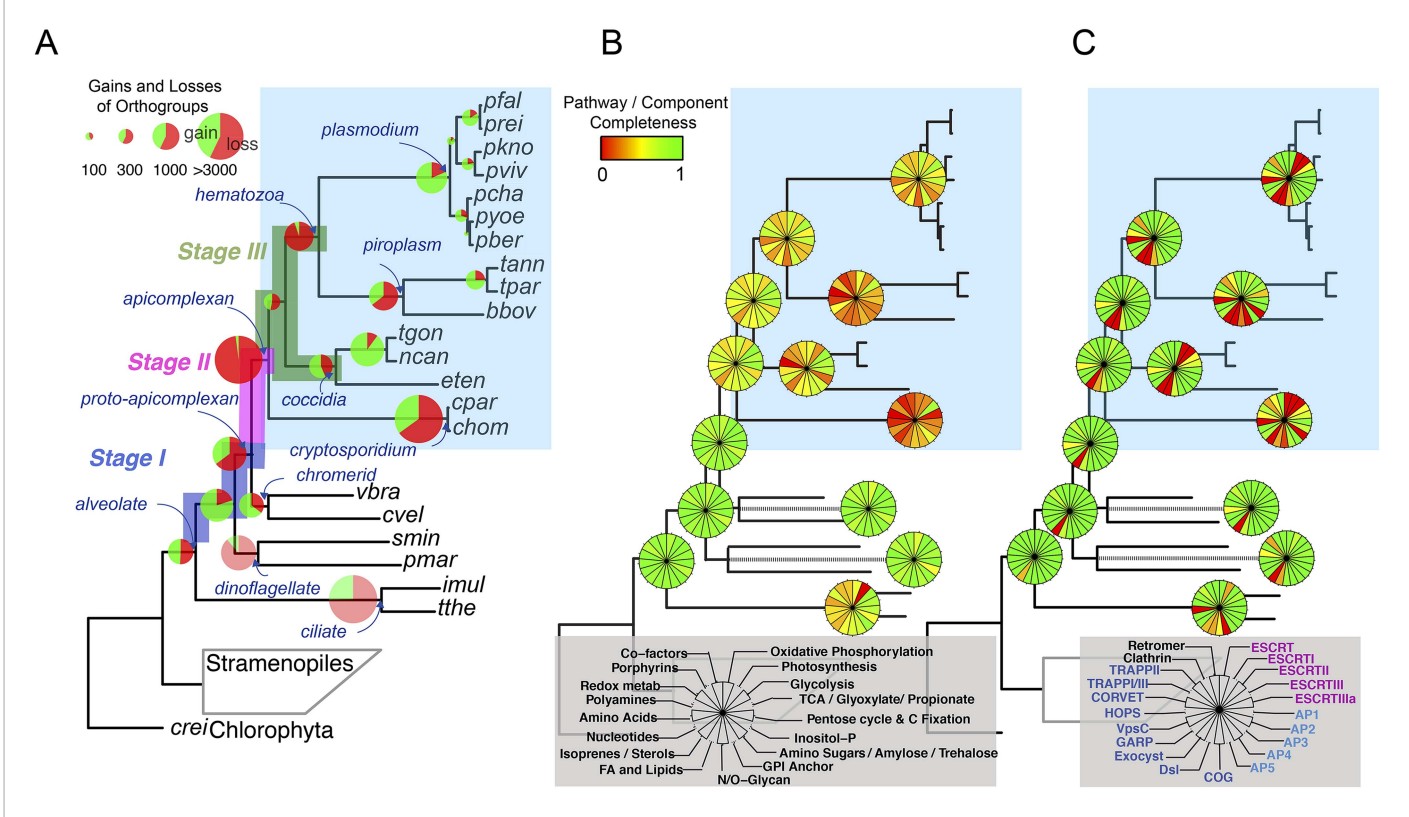

**Figure 2**. Gene content changes during apicomplexan evolution. (**A**) Gains and losses of orthogroups inferred based on Dollo parsimony (**Csuros, 2010**). Analysis based on a gene birth-and-death model provided similar results (**Figure 2—figure supplement 1A**). Stages I, II, and III (shown in blue, pink and green, respectively) represent groups of branches from the alveolate ancestor to apicomplexan lineage ancestors. Stage III could not be determined for Cryptosporidium lineage because of sparse taxon sampling. The area of a green or red section in a pie is proportional to the number of gained or lost orthogroups, respectively. (**B**, **C**) Overview of metabolic capabilities (**B**) and endomembrane components (**C**) in apicomplexan and chromerid ancestors. Gains and losses of enzymes and components were inferred, based on Dollo parsimony (**Csuros, 2010**). The pie charts are color-coded based on the fraction of enzymes or components present. Additional results from analysis of individual components and enzymes can be found in **Figure 2—figure supplements 2,3,4,5**, **Supplementary file 3**. Individual components and enzymes are listed in **Figure 2—source data 1, 2**. Similar analyses were performed for components encoding flagellar apparatus (**Figure 2—figure supplement 5B**).

The following source data and figure supplements are available for figure 2:

**Source data 1**. Distribution of enzymes based on KEGG.

**Source data 2**. Genes encoding subunits of the endomembrane trafficking system.

**Figure supplement 1**. Gene gains and losses across the hypothetical ancestors of the 26 species under study.

**Figure supplement 2**. Overview of chromerid Carbamoyl Phosphate Synthetase (CPS) and Fatty Acid Synthase I (FAS I).

**Figure supplement 3**. Summary of metabolic pathways based on KEGG Assignments.

**Figure supplement 4**. An overview of endomembrane trafficking components.

**Figure supplement 5**. Evolutionary history of genes encoding cytoskeleton across 26 species.

The following source data is available for figure 2s5:

**Figure supplement 5—source data 1**. Genes encoding components of the flagellar apparatus in the 26 species.

The inferred proto-apicomplexan ancestor, like present-day chromerids, possessed complete metabolic pathways for sugar metabolism, assimilation of nitrate and sulfite, and photosynthesis-related functions (*Figure 2B*, *Figure 2—figure supplement 3*, Appendix 2, and *Supplementary file 3*). Unlike in other photosynthetic algae, both *Chromera* and *Vitrella* initiate heme synthesis in the mitochondrion using aminolevulinate synthase (C4 pathway), which thus far has been found only in a few eukaryotic heterotrophs, such as *Euglena gracilis*, dinoflagellates, and apicomplexans (*Kořený et al., 2011*; *van Dooren et al., 2012*; *Danne et al., 2013*) (Appendix 2 and *Supplementary file 4*). Both chromerids and apicomplexans encode modular multi-domain fatty acid synthase I (FASI)/polyketide synthase enzymes and single-domain FASII components (*Figure 2—figure supplement 2A,B*). Treatment of *Chromera* with a FASII inhibitor triclosan showed decreased production of long chain fatty acids (*Figure 2—figure supplement 2C* and Appendix 2), suggesting that *Chromera* synthesizes short-chain saturated fatty acids using the FASI pathway, which are then elongated using the FASII pathway. This was previously demonstrated in *Toxoplasma*, an apicomplexan that possesses both FASI and FASII (*Mazumdar and Striepen, 2007*). Likely, the proto-apicomplexan ancestor was a phototrophic alga harboring characteristic metabolic features previously found only in apicomplexan parasites, especially with regard to plastid-associated metabolic functions (see above and other examples in Appendix 2) (*Kořený et al., 2011*; *van Dooren et al., 2012*; *Danne et al., 2013*).

Transition to an apicomplexan ancestor (Stage II) was accompanied by the loss of metabolic processes including photosynthesis and sterol biosynthesis (*Figure 2B* and *Figure 2—figure supplement 3*). The apicomplexan ancestor appeared to possess a significant complement of enzymes in various pathways (*Figure 2B*) (*Lim and McFadden, 2010*). The differentiation of apicomplexan lineages (Stage III) was accompanied by further lineage-specific losses: for example, loss of FASI in *Plasmodium* spp, loss of FASII in *Cryptosporidium* spp., which has also lost the apicoplast, and loss of enzymes mediating polyamine biosynthesis in all lineages except *Plasmodium* (*Figure 2B* and *Figure 2—figure supplement 3*). These support the notion that enzymes involved in cellular metabolism critical for free-living organisms were not completely lost during the transition to the apicomplexan ancestor, but were further lost during subsequent differentiation and host-adaptation of apicomplexan lineages.

The proto-apicomplexan had a nearly complete repertoire of the endomembrane trafficking complexes, and much of this repertoire persisted through to the apicomplexan ancestor (Stage II) (*Hager et al., 1999*; *Klinger et al., 2013a*) (*Figure 2C*, *Figure 2—figure supplement 4* and Appendix 3). Differentiation of apicomplexan lineages (Stage III) was accompanied by lineage-specific losses, for example, loss of the Endosomal Sorting Complex Required for Transport II (ESCRTII) in all lineages except in piroplasms, whereas some components were retained across all lineages, such as the retromer complex components and clathrin, both systems implicated in invasion processes (*Pieperhoff et al., 2013*; *Tomavo et al., 2013*) (*Figure 2C*, *Figure 2—figure supplement 4* and Appendix 3). These lineage-specific losses have led to diverse, reduced sets of endomembrane trafficking components in present-day apicomplexans (*Hager et al., 1999*; *Klinger et al., 2013a*). Some of these components that were present in chromerids were absent in specific apicomplexan lineages as well as in dinoflagellates and ciliates, further clarifying that these losses are independent, lineage-specific events rather than ancient, shared events.

All known components of flagella were present in the proto-apicomplexan ancestor (*Figure 2—figure supplement 5A,B*). Most of the components were retained in the apicomplexan ancestor (Stage II), but losses occurred as apicomplexan lineages differentiated (Stage III). Components of intraflagellar transport, which are typically essential for assembling flagella, were lost in the other lineages except in coccidians (*Figure 2—figure supplement 5A,B*). The basal body proteins, which support an organizing center for microtubules, were lost from piroplasms. Some striated fiber assemblin (SFA) proteins, typically associated with basal body rootlets, were maintained in all apicomplexan lineages including piroplasms (*Figure 2—figure supplement 5A,B,D*); their presence has been hailed as evidence that some flagellar-proteins are repurposed for new functions in apicomplexans (see below) (*Francia et al., 2012*).

In summary, one of the major events during apicomplexan evolution is progressive, continued loss of components important for free-living organisms. While Stage II was accompanied by a massive loss of such components including those implicated in photosynthesis, the apicomplexan ancestor still possessed many proteins, which were lost later during differentiation of lineages with diverse life strategies.

## Emergent features of apicomplexans

Evolution of present-day apicomplexan parasites was accompanied not only by gene losses as noted above (*Figure 2*) but also by gene gains. We sought to determine if genes gained at a particular stage

of apicomplexan evolution, as depicted by the gray violin in *Figure 3*, would be over-represented with those involved in parasitic processes such as intracellular invasion into and egress from host cells. For *Plasmodium falciparum* and *T. gondii*, we compiled three classes of protein-coding genes directly or indirectly involved in parasitic processes of apicomplexans based on in silico prediction or information from previous functional studies ('Materials and methods'). Extracellular proteins are secreted by the apicomplexans for various parasitic processes, for example, some of them are targeted to the host cytoplasm, nucleus, and plasma membrane to modulate parasite–host interactions

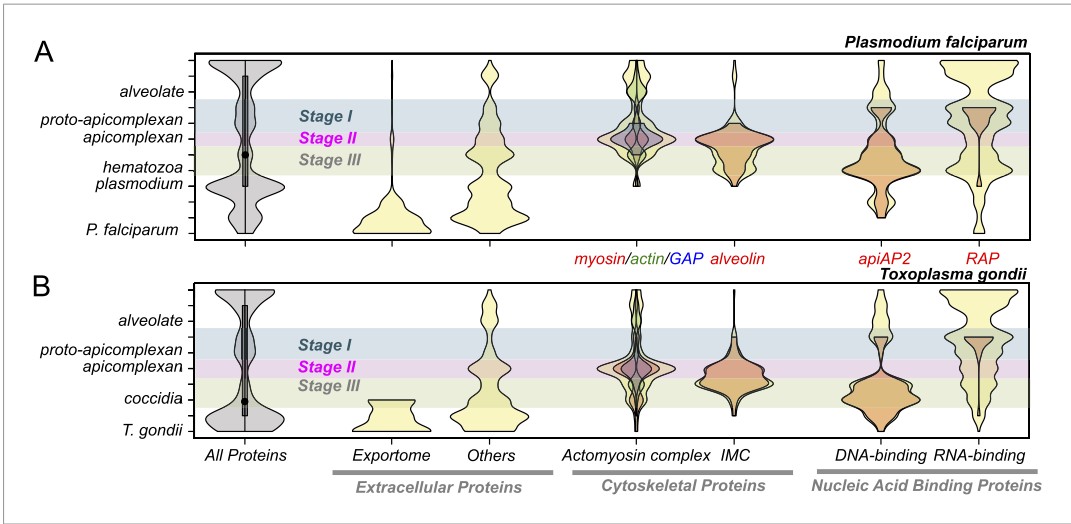

**Figure 3**. Evolutionary history of *Plasmodium falciparum* and *Toxoplasma gondii* genes. Violin plots showing distribution of evolutionary ages of genes (Y-axis: from species-specific (bottom) to deeply conserved (top)) in *P. falciparum* (**A**) and *T. gondii* (**B**). Evolutionary age of a gene is defined as the earliest node on the evolutionary path of the phylogenetic tree where homolog can be detected ('Materials and methods'). The horizontal thickness of a violin is proportional to the number of genes (gray) or the fraction of genes (yellow) in a functional category (X-axis) out of all with the same evolutionary age. Selected functional sub-categories are overlaid with red, green, or blue violin plots. The maximum width of each violin is scaled to be uniform across categories. Inner boxes in the gray violins indicate inter-quartile ranges and circles indicate medians. Colored shades along the X-axis indicate Stages I–III (*Figure 2*). Extracellular proteins include proteins targeted to host cytoplasm, nucleus, and plasma membrane ('exportome') and all other proteins, which are secreted or localized on the parasite surface ('others'). Cytoskeletal proteins include proteins associated with 'actomyosin motor complex' and 'IMC'. All extracellular and cytoskeletal proteins are listed in *Figure 3—source data 1, 2*. Nucleic acid-binding proteins are predicted in silico based on presence of DNA-binding domains (DBDs) and RNA-binding domains (RBDs). See 'Materials and methods' for details on how these genes are defined and compiled. Domain architectures of representative extracellular proteins in apicomplexans and chromerids are displayed as schematics in *Figure 3—figure supplement 4*. Sequence homology networks (*Figure 2—figure supplement 5E* and *Figure 3—figure supplements 1B, 2B, 3B*) and gene gains and losses on the phylogenetic tree (*Figure 3—figure supplements 1A, 2A, 3A*) provide complementary views on the evolutionary history of these genes.

The following source data and figure supplements are available for figure 3:

**Source data 1**. Genes encoding extracellular proteins in *P.falciparum* and *T. gondii*.

**Source data 2**. Genes encoding cytoskeletal components in the 26 species.

**Figure supplement 1**. Evolutionary history of apiAP2 genes.

**Figure supplement 2**. Evolutionary history of alveolins.

**Figure supplement 3**. Evolutionary history of RAP genes.

**Figure supplement 4**. Domain architectures of extracellular proteins in chromerids and apicomplexans.

(*Mundwiler-Pachlatko and Beck, 2013*; *Bougdour et al., 2014*). Cytoskeletal proteins provide structural support to the cell and also the molecular machinery for motility and intracellular invasion (*Baum et al., 2006*; *Soldati-Favre, 2008*). Proteins with DNA-binding domains (DBDs) or RNA-binding domains (RBDs) can regulate various molecular processes of apicomplexan parasites. Indeed, proteins with AP2 (apiAP2) DBD have been shown to act as genetic control switches for diverse apicomplexan processes (*Balaji et al., 2005*; *Campbell et al., 2010*; *Flueck et al., 2010*; *Radke et al., 2013*; *Sinha et al., 2014*; *Kaneko et al., 2015*).

Genes encoding extracellular proteins exported into the host environments were over-represented among those gained after Stage III (*Figure 3*), suggesting that adaptation to specific hosts was accompanied by expansion of extracellular proteins mediating host–parasite interactions (*Templeton et al., 2004a*; *Anantharaman et al., 2007*). Stage III was accompanied by gains of those encoding DBD proteins, mostly apiAP2 proteins (*Figure 3* and *Figure 3—figure supplement 1A,B*), suggesting extensive regulatory changes mediated by apiAP2 proteins during lineage differentiation. We note that losses of other canonical DBD proteins, for example, proteins with HSF_DNA-bind (Pfam: PF00447) domain during transition to apicomplexan ancestor (Stage II) and proteins with Tub (Pfam: PF01167) domain along the piroplasm lineage, contribute to further dominance of apiAP2 among the DBD proteins (*Figure 3—figure supplement 1C*). Stage II was accompanied by over-represented gains of various cytoskeletal components, including alveolins, those of the actomyosin complex (e.g., myosins) and glideosome-associated proteins with multiple membrane spans 1 and 3 (GAPM1 and GAPM3), suggesting that the molecular machinery powering gliding motility, which is essential for host cell invasion arose during evolution to apicomplexans (*Frenal et al., 2010*) (*Figure 3*, *Figure 3—figure supplement 2*, and Appendix 4). Gene gains during Stage I were over-represented by proteins with 'RBD abundant in Apicomplexans' (RAP, Pfam: PF08373) (*Lee and Hong, 2004*), many of which were conserved as one-to-one orthologs across descending lineages, suggesting development of evolutionarily conserved functions before apicomplexans and chromerids diverged (*Figure 3*, and *Figure 3—figure supplement 3*). Chromerid genomes encode many orthologs of apicomplexan cytoskeletal proteins (Appendix 4), including GAPM2, a member of an important protein family for apicomplexan cytoskeletal structure and gliding motility (*Bullen et al., 2009*), and the IMC sub-compartment protein family (ISP), implicated in establishing apical polarity and coordinating the unique cell cycle of apicomplexans (*Poulin et al., 2013*) (*Figure 2—figure supplement 5E*). These data suggest that some components existed in the free-living proto-apicomplexan ancestor and were subsequently repurposed for parasitic processes of apicomplexans.

The *Chromera* and *Vitrella* genomes encode many proteins that are specific to chromerids yet contain functional domains implicated in molecular processes of apicomplexan parasites. For example, there are chromerid-specific proteins with domain architectures similar to those in apicomplexan extracellular proteins, including those previously implicated in host interactions and described in apicomplexans only (*Figure 3—figure supplement 4* and Appendix 5, and *Supplementary file 5*). Presence of such chromerid proteins implies some commonality in extracellular recognition and cross-species interactions and this correlates well with the presumed associations with the coral holobiont (*Janouškovec et al., 2012*, *2013*; *Cumbo et al., 2013*). Importantly, chromerid genomes encode numerous apiAP2 proteins, more abundant than dinoflagellates, suggesting that they have expanded in the proto-apicomplexan ancestor after it split from dinoflagellates (*Figure 3—figure supplement 1D*). Many of the chromerid apiAP2 proteins belong to putative paralogous clusters, suggesting that their expansion was driven by gene duplication (*Figure 3—figure supplement 1D*; Appendix 6). Only a small subset of the apiAP2 proteins are shared across apicomplexans, suggesting that the large apiAP2 complement in the proto-apicomplexan ancestor has diversified independently in descending lineages (*Figure 3—figure supplement 1A*).

In summary, genes encoding critical components of the parasitic lifestyle of apicomplexans were gained at different stages of apicomplexan evolution, some implying subsequent specialization to particular host niches, but others suggesting early adaptations before committing to parasitic lifestyle. This is evident by chromerid orthologs of many such proteins, for example, RAP proteins and specialized cytoskeletal components. Further, chromerid genomes encode chromerid-specific proteins that are not detected as orthologs of apicomplexan proteins but still have functional domains implicated in parasitic processes in apicomplexans. Together, these data imply that a molecular transition had occurred in free-living ancestors of apicomplexans, providing a foundation for host–parasite interactions and further adaptation.

# Conserved gene expression programs in the proto-apicomplexan ancestor

*Chromera* and *Vitrella* genomes allowed us to reconstruct the gene content of the free-living ancestor of apicomplexans. To infer their putative functions using genome-wide gene expression information (*Hu et al., 2010*), we cultured *Chromera* under 36 different combinations of temperatures, iron and salt concentrations, and generated their gene expression profiles by RNA-seq (*Box et al., 2005*). In addition, we have obtained a publicly available growth perturbation data set for *P. falciparum* (*Hu et al., 2010*). There were 1918 orthogroups shared between the two species. We identified pairs of orthogroups that are co-expressed, that is, showing similar expression patterns across the various conditions, in both species ('Materials and methods') (*Figure 4—figure supplement 1A*). Such an orthogroup pair, that is, those with conserved co-expression between the two species, would include candidate genes that have been co-regulated together during apicomplexan evolution, from the free-living ancestor to present-day parasites due to conserved functions. This approach, successfully utilized by several studies in the past (*Stuart et al., 2003*; *Mutwil et al., 2011*; *Gerstein et al., 2014*), led to the following two observations in this study.

Many RAP genes appeared during Stage I and have been conserved across the descending phyla (*Figure 3* and *Figure 3—figure supplement 3*), but their precise cellular roles are unknown. For 11 out of 12 orthogroups with RAP domains, co-expressed orthogroups overlapped significantly (Fisher's exact test, $p < 0.05$) between *P. falciparum* and *Chromera*, suggesting involvement of RAP proteins in cellular processes evolutionarily conserved across apicomplexans and chromerids (*Figure 4A*). RAP and their co-expressed orthogroups encode proteins with putative mitochondrial import signals more often than expected by chance in *Chromera* and *P. falciparum* (Fisher's exact test, $p < 0.05$) (*Figure 4B*), and also in other apicomplexans and chromerids (*Figure 4—figure supplement 1B*). We have randomly chosen three *Toxoplasma* RAP genes with predicted mitochondrial localization signals (*Supplementary file 6*) and confirmed experimentally by 3′ endogenous gene-tagging with reporter epitopes that all three are localized to the organelle (*Figure 4C*). Some of the orthogroups co-expressed with orthogroups containing RAP domains encode protein products predicted to be metabolic enzymes, implying possible involvement of RAPs in mitochondrial metabolism (*Figure 4—figure supplement 1C*). Consistent with this, the *Cryptosporidium* lineage that has a highly reduced mitochondrion lacking both the genome and most canonical metabolic pathways (*Abrahamsen et al., 2004*; *Xu et al., 2004*) is the only apicomplexan group to have also lost its RAP repertoire (*Figure 4—figure supplement 1D*). Loss of RAPs along with a set of mitochondrial functions in this lineage is consistent with a mitochondrial role for RAPs. We speculate that the free-living proto-apicomplexan ancestor possessed within its mitochondrion a regulatory process mediated by RNA-binding activities of the RAP proteins, which has been retained by the extant apicomplexans and chromerids.

As discussed earlier, the proto-apicomplexan ancestor appears to have possessed genes implicated in invasion processes of present-day apicomplexans (*Figure 3*). Among the 1918 orthogroups, we identified 80 orthogroups comprising genes functionally annotated as implicated in invasion processes. The frequency of co-expression amongst them in the free-living *Chromera* was significantly higher than expected by chance ($p < 0.0005$), suggesting pre-existing functional relationships before transitioning to parasites (*Figure 4D*). We identified several modules or groups of co-expressed orthogroups (*Figure 4E*). In one of the co-expression modules (numbered 1 in *Figures 4E*), 9 out of 10 orthogroups are co-expressed with a gene encoding SFA (Cvel_872), a key protein for organizing the basal bodies of the flagellar apparatus in algae and the apical complexes in apicomplexans (*Kawase et al., 2007*; *Francia et al., 2012*) (*Figure 4F*). We note that SFAs are the only flagellar components found in all apicomplexans tested (*Figure 2—figure supplement 5A*). Also in this module, for 9 out of 10 orthogroups, their co-expressed orthogroups in *Chromera* overlapped significantly with those in *P. falciparum* (Fisher's exact test, $p < 0.05$), indicating that their regulatory programs have been evolutionarily conserved (*Figure 4G*). This module include various types of genes implicated in host cell invasion processes of apicomplexans such as genes encoding rhoptry protein ROP9, apical sushi protein ASP, and gliding motility components GAP40 and GAPM2. The apical complex has been postulated to have emerged from the flagellar apparatus and associated cellular transport systems in free-living algae, based on ultrastructural evidence (*Okamoto and Keeling, 2014*; *Portman et al., 2014*). These results suggest that, in the free-living ancestor, some of the genes

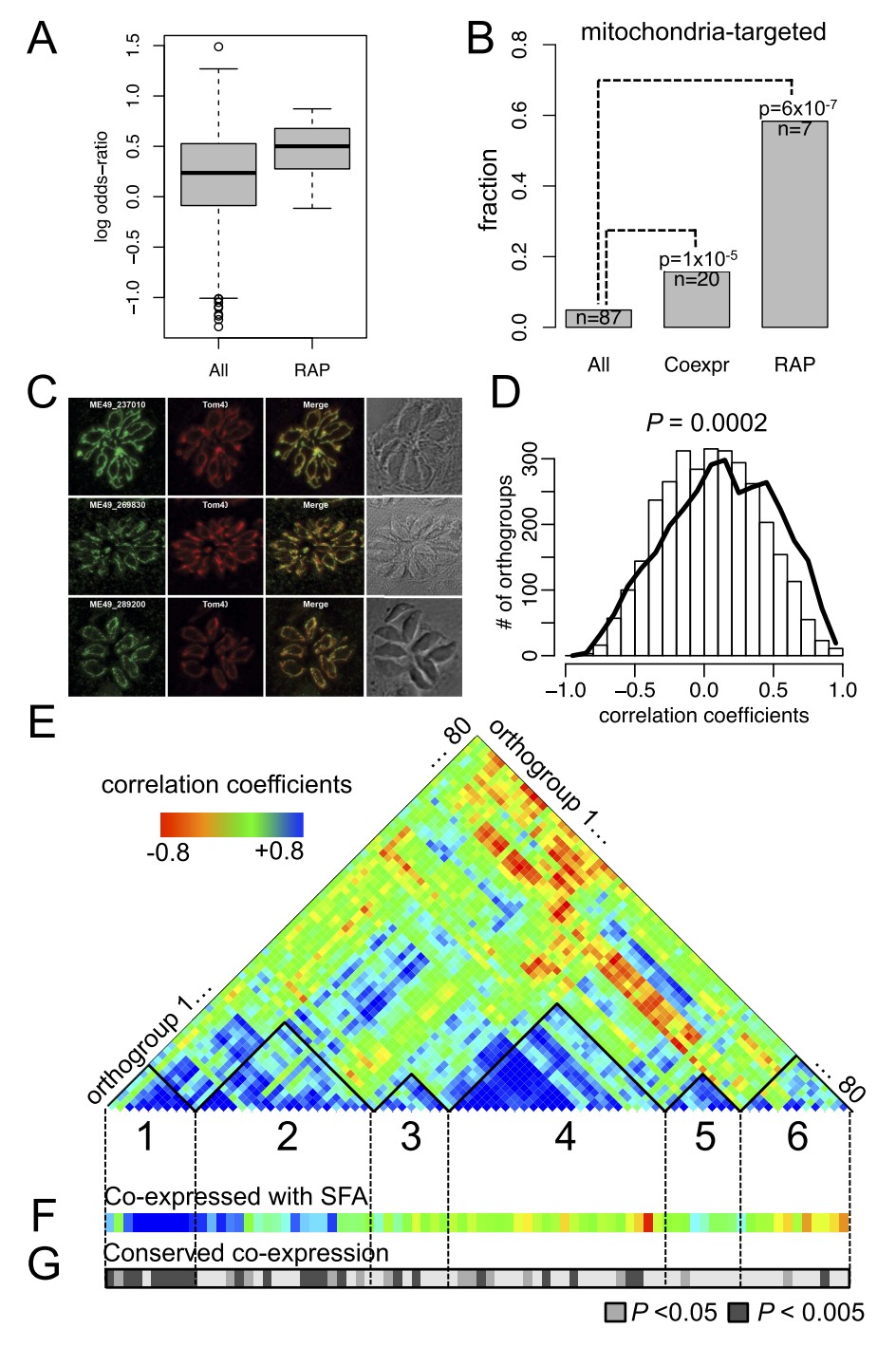

**Figure 4**. Conserved transcriptional programs in apicomplexans and chromerids. (**A**) Boxplot showing the extent of evolutionary conservation of transcriptional programs for all orthogroups or those with RAP domains. X-axis: 'All' (all orthogroups excluding RAP); 'RAP' (orthogroups with RAP domains). Y-axis: log-transformed odds-ratio, representing, for each orthogroup, the degree of overlap between its co-expressed orthogroups in Chromera and those in *P. falciparum*. (**B**) Bar chart showing the fraction of orthogroups (Y-axis) predicted to be targeted to mitochondria in both species ('Materials and methods'). The number of genes are displayed below each bar. X-axis: 'All' (all orthogroups excluding the other two categories); 'Coexpr' (orthogroups co-expressed with RAP in both species); 'RAP' (orthogroups with RAP domains). The fractions in 'Coexpr' and 'RAP' groups were compared against the fraction in 'All', and p-values based Fisher's exact test are displayed above the bar. Files deposited in European Nucleotide Archive are listed in *Figure 4—source data 1* with corresponding conditions. (**C**) Sub-cellular
*Figure 4. continued on next page*

*Figure 4. Continued*

localization of RAP proteins encoded by TGME49_237010, TGME49_269830, and TGME49_289200 was tested in *T. gondii* by 3′ tagging of the endogenous genes with the coding sequence for the hemagglutinin epitope, together with a mitochondrial marker Tom40. See *Supplementary file 6* for details of the localization predictions. (**D**) Distributions of Spearman's rank correlation coefficients of gene expression between all possible pairs from the 80 orthogroups implicated in invasion processes in apicomplexans (black outline) were compared against those from 80 randomly selected ones (histogram). The p value indicates statistical significance of the difference based on 10,000 random samplings. The 80 orthogroups and corresponding genes in *Chromera* and *P. falciparum* are listed in *Figure 4—source data 2*. (**E**) Heatmap showing a matrix of correlation coefficients amongst the 80 orthogroups. Based on a hierarchical clustering, we classified them into six co-expression modules, labeled as numeral 1–6. (**F**) Heatmap showing correlation coefficients with striated fiber assemblin (SFA) (Cvel_872). The color scheme is the same as in (**E**). (**G**) Heatmap indicating statistical significance of conserved transcriptional program, that is, the odds-ratio as defined in (**A**) (Fisher's exact test, $p < 0.05$ (gray); $p < 0.005$ (black)).

The following source data and figure supplement are available for figure 4:

**Source data 1**. RNA-seq libraries of *Chromera velia* under various growth conditions.

**Source data 2**. List of genes implicated in invasion processes in apicomplexans.

**Source data 3**. Evolutionary conservation of 12 orthogroups with RAP domains (for 'RAP' category in *Figure 4A*).

**Figure supplement 1**. Mitochondrial targeting of RAP and its putative role in mitochondrial metabolism.

---

implicated in the invasion process of present-day apicomplexans were functionally associated with those implicated in flagellar motility, providing the much-needed genetic evidence for the postulate. We speculate that a group of functionally related proteins associated with the flagellar apparatus was repurposed as a module of the apical complex and became a foundation for the invasion machinery.

## Conclusion

Analysis of *Chromera* and *Vitrella* genomes has enabled insights into how apicomplexan parasites have evolved from free-living ancestors. The transition to parasitism was accompanied by massive genomic loss that continued as its descendants became specialized intracellular parasites infecting diverse hosts. The genome of free-living photosynthetic ancestors encodes many component proteins previously assumed to be restricted to the parasitic apicomplexan lineages. Such pre-existing components, including those of what would later become part of the invasion machinery, were co-opted during evolution to facilitate a successful parasitic lifestyle in multiple hosts. The genome of the proto-apicomplexan ancestor served as a molecular blueprint for evolution of the most successful group of eukaryotic parasites known to date.

## Data access

Sequencing data have been deposited in the European Bioinformatics Institute under the European Nucleotide Archive (ENA) sample accession number ERP006228 for *C. velia* and ERP006229 for *V. brassicaformis* for all DNA- and RNA-seq experiments. The assembly and the annotations were submitted under accession numbers CDMZ01000001-CDMZ01005953 for *C. velia* and CDMY01000001-CDMY01001064 for *V. brassicaformis*. Some of the *Vitrella* DNA-seq experiments were done at Broad Institute and are deposited at Short Read Archive under accession numbers SRX152523 and SRX152525. The annotations and assemblies can be viewed and queried in EupathDB (http://cryptodb.org/cryptodb/).

## Materials and methods

### DNA preparation and sequencing

Genomic DNA of *C. velia* CCMP2878 (subsequently referred to as *Chromera*) and *V. brassicaformis* CCMP3155 (subsequently referred to as *Vitrella*) was extracted and then sheared into short fragment

size libraries (300–500 base pair (bp)) and large fragment size libraries (3–8 kbp fragments) by focused-ultrasonication (Covaris Inc., Woburn, USA). The last 3–8 kb libraries were prepared following Nextera mate pair protocol, following manufacturer's instructions. We used three different methods to generate the library: the Illumina (Illumina, San Diego, CA) TruSeq DNA protocol LT Sample Prep Kit (catalog no. #FC-121-2001), an amplification-free method (*Kozarewa et al., 2009*) (TruSeq DNA PCR-Free LT Sample Preparation Kit catalog no. #FC-121-3001) and the Illumina Nextera Mate Pair Sample Preparation Kit (catalog no. #FC-132-1001). The libraries were sequenced on an Illumina HiSeq2000 platform following the manufacturers standard cluster generation and sequencing protocols (*Bentley et al., 2008*; *Quail et al., 2012*). Image analysis, base-calling, and quality filtering were processed by Illumina software.

## RNA preparation and sequencing

For isolation of RNAs, *Chromera* and *Vitrella* were grown under standard culture conditions (*Oborník et al., 2012*). Total RNA was extracted from the cells using TRIzol. The polyA+ RNA fraction was selected using oligo(dT) beads, and RNA-seq libraries were prepared using TruSeq RNA Sample Prep kit (catalog no. FC-122-1001). Strand-specific RNA-seq libraries were prepared using TruSeq Stranded mRNA LT Sample Prep Kit (catalog no. RS-122-2101) and sequenced as paired-end (2 x 100 bp) reads on a HiSeq2000 platform.

We performed additional RNA sequencing of *Chromera* subject to various environmental perturbations, to construct a global gene expression network based on transcriptomes under various perturbation conditions during in vitro growth. *Chromera* cultures were exposed to a combination of stresses (*Figure 4—figure supplement 1C*). First, six different media were prepared from the combinations of salt concentration (16.7 g/l, 33.3 g/l, 66.6 g/l) and iron deficiency by chelation (*Sutak et al., 2010*). After seeding, the cultures were maintained in the normal temperature and light condition for eleven days (*Oborník et al., 2011*). After randomization, the cultures were incubated at 26°C, 37°C, or 14°C for 0 (control), 0.5, or 2 hr. There were two biological replicates of each, in total 66 flasks of the cultures. Then, the cultures were processed with centrifugation at 3500 RPM for 15 min at 4°C to precipitate the cells. Total RNA was extracted from the 66 cultures after the treatments using Norgen RNA Extraction kit based on manufacturer's protocol (Norgen Biotek Corporation, Canada). RNA quality was assessed using Bioanalyzer 2100 (Agilent Technologies, Santa Clara, CA). RNA concentration was determined with a Qubit (Invitrogen, Carlsbad, CA). Strand-specific RNA-seq libraries were prepared from extracted high-quality RNAs (RIN ≥8.0 as measured on an Agilent Bioanalyser 2100) using the Illumina TrueSeq LT stranded RNA sample kit according to manufacturer's instructions. Prior to cluster generation, concentration and size of libraries were assayed using the Agilent DNA1000 kit. Libraries from all samples were sequenced as single-end (1 x 50 bp) reads on the Illumina HiSeq 2000. The RNA-seq reads were aligned to the reference genome using tophat (version 2.0.8, default parameters) and cufflinks (version 2-1.0.2, default parameters) (*Trapnell et al., 2012*). The FPKM values were *log2* normalized with an offset of 1 and were further corrected for different distributions across the samples using the quantile normalization method (*Bolstad et al., 2003*).

## Genome assembly

For *Vitrella*, the reads were corrected and assembled followed by several base correction, scaffolding and gap filling steps as briefly described below. As first step, the short insert libraries were corrected with SGA (*Simpson and Durbin, 2012*) (version 0.9.19). The corrected reads were assembled with velvet (*Zerbino and Birney, 2008*) (version 1.2.08). Iterating through different parameter settings, we choose a k-mer of 75 bp as the best parameter set. The resulting scaffolds (larger than 1 kb) were further scaffolded with SSPACE (*Boetzer et al., 2011*) using first the Illumina library (insert = 550 bp) and larger insert (1 kb) Illumina library reads. Sequencing gaps were closed with Gapfiller (*Boetzer and Pirovano, 2012*) (version 1.1.1) with two iterations, using the bowtie mapping option and PCR-Free libraries. Base pair call errors were corrected in three iterations of ICORN (*Otto et al., 2010*), using the amplification-free library. Furthermore, sequencing gaps were closed, using IMAGE (*Tsai et al., 2010*) with the amplification-free library. The assembly was quality-controlled using REAPR (*Hunt et al., 2013*), breaking the contigs at possible miss-assemblies, using the mate pair libraries. This was followed by another scaffolding step. We systematically removed 620 scaffolds containing 25.65 Mb representing the bacterial contamination. The *Vitrella* CCMP3155 assembly contains 72.7

Mb (including 931,689 N's) in 1064 scaffolds (ENA accession numbers CDMY01000001-CDMY01001064). The scaffolds were constructed from 4177 contigs.

For *Chromera*, the assembly pipeline and the algorithms used were the same as *Vitrella*, but due to the larger size, higher amount of low-complexity regions, and difficulties in generating high-quality large insert size libraries, additional steps were included to the assembly process. First, the reads of the PCR-Free library were corrected with SGA (*Simpson and Durbin, 2012*) and then assembled with velvet and using a k-mer of 71 (version August 2011). Next, the contigs were scaffolded, gapfilled, and corrected with ICORN, as described earlier. We mapped the reads of all large insert size libraries using SMALT (ftp://ftp.sanger.ac.uk/pub/resources/software/smalt/). We excluded scaffolds smaller than 1 kb. Different iterations with SSPACE were undertaken and the assembly was quality-checked with REAPR. After scaffolding, gapfiller and IMAGE were run as above, followed by ICORN. The 1725 scaffolds (spanning 16.02 Mb) representing bacterial contamination were removed. The final assembly of *Chromera* CCMP2878 contains 193.66 Mb (including 582,995 N's) in 5953 scaffolds (ENA accession numbers CDMZ01000001-CDMZ01005953). The scaffolds are constructed from 13,987 contigs.

## Gene prediction

We used Augustus (*Stanke et al., 2006*) (version 2.5.5) for gene prediction. We manually curated 716 and 245 gene models for *Chromera* and *Vitrella*, respectively, using BLAST similarity-based approaches, and we also generated automated gene models using Cufflinks (*Trapnell et al., 2012*) from RNA-seq data sets, in order to use them as a 'training gene model set' for Augustus prediction. The strand-specific RNA-Seq, mapped with TopHat2 (*Kim et al., 2013*), was used as evidence in Augustus for intron evidence.

In summary, from the *Chromera* and *Vitrella* genome, we ab initio predicted 30,478 and 23,503 protein-coding genes, respectively, of which 18,829 and 18,240 were detected as being expressed from RNA-seq evidence as poly A+ transcripts (*Supplementary file 1*). Excluding putative TEs, 26,112 and 22,817 genes were predicted as protein-coding genes in *Chromera* and *Vitrella*. We annotated partial genes, when a gene probably spans more than one scaffold, located at the borders of a scaffold. We demarcated and annotated as pseudo genes if they contain in frame stop codons. We flagged gene models as transposon elements, if they overlap with the predicted TE regions and had no more than three and two intron for *Chromera* and *Vitrella*, respectively. To annotate untranslated regions (UTRs) of the predicted protein-coding genes, we used CRAIG (*Bernal et al., 2007*) with default parameters with mapping of the RNA-Seq data as computed by GSNAP (*Wu and Nacu, 2010*) (version 2013-08-19, default parameters). The annotation of both genomes has the ENA accession numbers CDMZ01000001-CDMZ01005953 and CDMY01000001-CDMY01001064 and is also available in EuPathDB (*Aurrecoechea et al., 2013*).

## Functional annotations

The predicted genes were assigned putative functions based on BLASTP (E value $<10^{-6}$) matches against UNIPROT (version March 2012). The predicted protein products were assigned protein domains using *hmmsearch* (HMMER 3.1b1, May 2013) for Pfam A v26.0. Statistical threshold defined by the Pfam (*Finn et al., 2014*) database was used. We aligned AP2 sequences in apicomplexan species based on PfamA AP2 (PF00847), and built apicomplexan-specific AP2 (apiAP2) hidden Markov model (HMM), and scanned the predicted protein-coding genes for apiAP2 domains; we annotated api-AP2 DNA-binding transcription factor genes with both domain and sequence E values to be less than $10^{-3}$. The following Pfam RBDs were used to define RNA-binding proteins: 'CAT_RBD', 'dsRNA_bind', 'S1', 'DEAD', 'KH_1', 'KH_2', 'KH_3', 'KH_4', 'KH_5', 'RRM_1', 'RRM_2', 'RRM_3', 'RRM_4', 'RRM_5', 'RRM_6', 'SET', 'PUF', and 'RAP'. The list of DBDs was downloaded from a database of DBDs (*Wilson et al., 2008*). Transmembrane domains and signal peptides were assigned with the tools TMHMM 2.0 (*Krogh et al., 2001*) and signalP 4.0 (*Petersen et al., 2011*), respectively, with default parameters.

We collected several categories of genes implicated in parasitic processes in apicomplexans for two archetypal apicomplexan parasites, *Toxoplasma* and *Plasmodium*. We primarily obtained annotations from PlasmoDB (*Bahl et al., 2003*) and ToxoDB (*Gajria et al., 2008*). Information for sub-cellular localization of genes is obtained from GeneDB (*Logan-Klumpler et al., 2012*) and ApiLoc, a database of published protein sub-cellular localization for apicomplexan species

(http://apiloc.biochem.unimelb.edu.au/apiloc/apiloc). Some putative parasite genes were inferred based on orthology by OrthoMCL clustering (*Li et al., 2003*) with closely related species with results from functional studies. We performed exhaustive literature searches to manually curate individual genes, to define rules for in silico searches across the proteomes of this study, and to categorize the identified genes based on their localization and function. The categories of parasite genes are defined as follows.

## Cytoskeleton

The cytoskeleton of an organism provides the necessary structural framework for the maintenance of cell shape and integrity. We compiled two groups of cytoskeletal proteins, IMC associated proteins and actomyosin complex. First, IMC associated proteins, comprises alveolin proteins, a membrane occupation and recognition nexus protein (MORN), which associate with IMC and spindle poles and are indispensible for asexual and sexual development (*Ferguson et al., 2008*). IMC sub-compartment proteins (ISPs) are critical for establishing apical polarity in the parasite (*Poulin et al., 2013*). Second, components of actomyosin motor complex, which powers the characteristic gliding motility (*Soldati-Favre, 2008*), comprises actin, myosin, tubulin, gliding associated proteins (GAPs), aldolase, and various actin-regulatory proteins, which will assist actin in the process of quick polymerization–depolymerization cycles between F-actin and G-actin during this process. Examples of actin-regulatory proteins are Arp2/3 complex and formins (FH2) for nucleation; F-actin capping for filament regulation; coronin for cross-linking/bundling and profilin, CAP, cofilin/ADF and gelsolin for monomer treadmilling (*Baum et al., 2006*).

## Extracellular proteins

Extracellular proteins are defined as parasite proteins, which are localized either on the surface or secreted off the parasite. They are released in a concerted manner to ensure successful adhesion to the surface, entry into the host cell, multiplication, and escape. Extracellular proteins can be categorized as (1) 'exportome' are proteins translocated to the host cytoplasm, membranes, and nucleus crossing the boundary membrane parasitophorous vacuole (PV); and (2) 'others', which stay on the parasite surface or released from the parasite, but not into the host intracellular space. The exportome genes are released mostly from the parasite's secretary organelles such as rhoptries and dense granules (*Ravindran and Boothroyd, 2008*; *Treeck et al., 2011*; *Mundwiler-Pachlatko and Beck, 2013*; *Bougdour et al., 2014*). Some of these genes possess host targeting or also known as the Plasmodium export element (PEXEL). Many PEXEL-negative proteins have been identified too (*Hsiao et al., 2013*; *Mundwiler-Pachlatko and Beck, 2013*). These genes are sorted and targeted through a specialized structure known as Maurer's cleft formed in the host cytoplasm (*Mundwiler-Pachlatko and Beck, 2013*). These genes are mostly kinases, proteases, and surface molecules, which modulate the host and hijack the host machinery in favor of parasitic growth and host immune evasion (*Treeck et al., 2011*; *Li et al., 2012*; *Bougdour et al., 2014*). The 'other' extracellular proteins consist of surface antigens (e.g., MSPs), SERAs, TRAPs, AMA-1, microneme proteins, ROPs and RONs etc.

## TEs

Repeat annotation was done by using the REPET pipeline (*Flutre et al., 2011*) and LTR finder (*Xu and Wang, 2007*). The overall pipeline comprises of two steps: de novo detection and classification. In the first step, the scaffolds are split into smaller batches (~1000 batches of 200 kb each). These genomic fragments were aligned against each other to detect the HSPs (High-scoring pairs) using BLASTER (*Quesneville et al., 2003*). HSPs are then clustered using a combination of three methods such as GROUPER (*Quesneville et al., 2003*), RECON (*Bao and Eddy, 2002*), and PILER (*Edgar and Myers, 2005*). Structure-based LTR retrotransposons (RTs) detection tools such as LTRharvest (*Ellinghaus et al., 2008*) and LTR finder, which are based on 100–1000 bp long terminal repeats with a 1 kb–15 kb separation and target site duplication site at vicinity of 60 bp to the two terminal repeats. These LTRs detected are clustered using BlastClust. Multiple sequence alignment of each cluster was performed using MUSCLE (Edgar, 2004). Each cluster aligned was searched against Repbase (*Jurka et al., 2005*) using BLASTER (*Quesneville et al., 2003*) and HMMER (*Johnson et al., 2010*). A consensus feature was detected for each aligned cluster. Further PASTEC (*Flutre et al., 2011*), which is based on the Wicker classification, was used for consensus classification.

The repeats were annotated as follows. The genomic chunks were randomized and HSPs were detected using BLASTER (*Quesneville et al., 2003*), CENSOR (*Jurka et al., 1996*), and RepeatMasker (*Tempel, 2012*). These HSPs were filtered and combined. Again, full-length genomic scaffolds were compared to Repbase using MATCHER. Satellite and simple repeats were detected using the mreps (*Kolpakov et al., 2003*), TRF (*Benson, 1999*), RMSSR (RepeatMasker). Finally, a long-join procedure was followed to combine the nested repeats. The whole annotation was exported to a genome-browser readable GFF3 file.

## Clustering homologous genes

OrthoMCL 2.0 (*Li et al., 2003*) was used with a default inflation parameter (I = 1.5) (*Chen et al., 2006*) to generate groups of homologous genes (defined as orthogroups), which could have homologs from different species (putative orthologs) or from the same species (putative paralogs from gene duplications). For some genes of high interest, we manually inspected the alignments of the protein sequences within the orthogroup, which were done with MAFFT (*Katoh and Standley, 2013*). We assigned Pfam domains to an orthogroup if more than half of the genes in an orthogroup were assigned the Pfam domains.

## Sub-cellular localization prediction

There are several tools available for a general eukaryotic sub-cellular localization prediction (*Du et al., 2011*), but they are not applicable to alveolates due to its unique chloroplast membrane arising from secondary endosymbiosis. Therefore, HECTAR (*Gschloessl et al., 2008*), which was developed for the bipartite sub-cellular prediction, was used. There is no stand-alone version of HECTAR, and the online version allows only one sequence at a time. We implemented a modified HECTAR algorithm as a PERL script for batch prediction of the whole proteomes. Each protein sequence was predicted for signal sequence using SignalP 3.0 (*Bendtsen et al., 2004*), the signal sequence is cleaved, and the remaining amino acid sequence was used as input for the transit peptide prediction by TargetP (*Emanuelsson et al., 2000*). Sequences with both signal peptide and the transit peptide (either chloroplast or mitochondria) are predicted to be in the chloroplast. Sequences without the signal peptide but with the transit peptide (either chloroplast or mitochondria) are predicted to be in mitochondria. Sequences with signal peptide, without transit peptide, and predicted by TargetP to be secretory are classified as secretory proteins.

For the RAP proteins, we tested the validity of our sub-cellular localization prediction in two ways. First, we compared our in-house algorithm with other published tools: TargetP (*Emanuelsson et al., 2000*), MitoProt2 (*Claros and Vincens, 1996*), iPSORT (*Bannai et al., 2002*), and PredSL (*Petsalaki et al., 2006*) (*Supplementary file 6*, only mitochondrial prediction is shown). We found that our mitochondrial prediction for RAP genes is in concordance with other methods. Second, we experimentally verified mitochondrial localization in *T. gondii* by 3′ tagging of the endogenous genes with the coding sequence for the hemagglutinin epitope for three RAP proteins that were predicted to target to mitochondria with high probability.

## Statistical analysis

A statistical environment software R was used for most of the analyses and generating parts of figures. An R package *vioplot* was used to generate the violin plot (*Hintze and Nelson, 1998*). A ward algorithm on the distance matrix based on (1- correlation coefficients) in an R function *hclust* was used for all hierarchical clustering of gene expression patterns unless noted otherwise.

## Evolutionary analysis

We compiled the reference proteomes of 26 alveolate and stramenopile species (*Figure 1—source data 2*) from public databases such as EupathDB (*Aurrecoechea et al., 2013*) and NCBI Genome database (http://www.ncbi.nlm.nih.gov/genome/).

We generated a phylogenetic species tree using a data set composed of 101 one-to-one orthologs across the 26 species (see *Figure 1—source data 1* for gene IDs). Amino acid sequences were aligned using MAFFT (*Katoh and Standley, 2013*), highly variable sites were edited by trimAL (*Capella-Gutierrez et al., 2009*) and after manual inspection. The resulting alignment of 33,997 amino acid positions was used to construct trees by a maximum likelihood

(ML) method and Bayesian inference. The ML tree was computed using RAxML 8.1.16 by gamma corrected LG4X model (*Stamatakis, 2014*; *Le et al., 2012*). Robustness of the tree was estimated by bootstrap analysis in 1000 replicates. Bayesian tree was constructed by PhyloBayes (*Lartillot and Philippe, 2004*) using two-infinite mixture model CAT-GTR as implemented in PhyloBayes 3.3f. Two independent chains were run until they converged (i.e., maximum observed discrepancy was lower than 0.2), and the effective number of model parameters was at least 100 after the first 1/5 generation was omitted from topology and posterior probability inference. All clades in the tree were supported with posterior probability 1.00 and 100% bootstraps, except for one node, which representing the common ancestor of human *Plasmodium* spp. was supported by 99% bootstrap.

We performed the gene gain and loss analysis based on Dollo parsimony using Count software (*Csuros, 2010*). This approach allows reconstructing gene contents at observed species and at hypothetical ancestors, and gene gains and losses at branching points. The Dollo parsimony strictly prohibits multiple gains of genes. To test for validity of this assumption, we repeated analyses based on parsimony settings allowing multiple gene gains or on a phylogenetic birth-and-death model (*Csuros, 2010*) and reached the same conclusion (*Figure 2—figure supplement 1*). We have also repeated the analysis using Wagner's parsimony, allowing multiple gains per tree with gain penalty of 2 or greater, and obtained similar results (data not shown). For the analysis of metabolic enzymes, endomembrane trafficking system components, and flagellar apparatus components, the ancestral presence was inferred based on Dollo parsimony from the presence of components in the observed species. For the endomembrane trafficking component analysis, we assumed that the last common ancestor had a complete repertoire of the components.

We have inferred the evolutionary age of *P. falciparum* and *T. gondii* genes as the early node on the phylogenetic tree where the most distant species have genes with significant sequence homology (reciprocal BLASTP E value $<10^{-10}$ and clustering with OrthoMCL).

## Comparison of gene expression network between *Chromera velia* and *Plasmodium falciparum*

We studied if orthologs of *Chromera* and *P. falciparum* show similar gene expression changes to physiologically equivalent growth conditions. Identifying equivalent conditions is difficult as the two species have completely different lifestyles and live in different environments. Instead, we tested if a given gene and its ortholog would show correlated expression patterns with the same set of genes (and orthologs), allowing a way to compare gene expression behavior measured under different conditions. To uncover gene-to-gene co-expression relationships, the organisms from whom transcriptomes are sampled must be exposed to various growth conditions. This approach has been successfully used in other eukaryotes (*Stuart et al., 2003*; *Hu et al., 2010*; *Mutwil et al., 2011*). For *Chromera*, we generated RNA-seq-based transcriptome under combinations of varying salt concentrations, iron concentrations, and temperature changes, resulting in 36 unique combinations (see 'Materials and methods' and *Figure 4—figure supplement 1C*). For *P. falciparum*, we obtained previously published microarray-based gene expression data sets of 144 unique conditions from 23 time series, representing stresses from various growth-inhibiting compounds (*Hu et al., 2010*). It has been shown that gene expression data generated using different molecular platforms are reproducible and accurate enough for cross-platform comparisons (*Woo et al., 2004*). Based on each data set, we calculated Spearman correlation coefficients *rho* between all possible pairs from the 1918 orthogroups shared between *Chromera* and *P. falciparum* (1918 × 1918 matrix). We also calculated a 1918 × 1918 weighted adjacency matrix using CLR algorithm (*Faith et al., 2007*) as implemented in an R package *minet* (with parameters of method = 'clr', estimator = 'mi.shrink', and disc = 'equalfreq') (*Meyer et al., 2008*). Expression level of multiple genes in a given orthogroup was averaged. To rule out any potential systematic biases associated with averaging expression levels of homologous, yet distinct genes, we repeated some of the analyses with 1560 orthogroups that have one-to-one orthologs between the two species and reached the same conclusions (data not shown). A pair of genes (or orthogroup) were determined as co-expressed if the Spearman's correlation coefficient *rho* is greater than 0.3 and if the value from the weighted adjacency matrix of the network is greater than 0.01. We calculated an odds-ratio to measure the

extent of conservation of co-expressed genes: (# of genes co-expressed in both species) × (# of genes co-expressed in none of the species)/([# of genes co-expressed in *P. falciparum* only] × [# of genes co-expressed in *C. velia* only]), and Fisher's exact test was used to assess the statistical significance. For calculation of the odds-ratios, co-expression was determined based on correlation coefficient to minimize count granularity in the two-by-two table.

## Acknowledgements

We thank the KAUST Bioscience Core Laboratory personnel for sequencing specific Illumina libraries used in this project, KAUST Computational Bioscience Research Center for providing computing resources, Gordon Langsley (Institut Cochin, Inserm U1016, Paris) and Anthony Holder (The Francis Crick Institute, London) for comments on the manuscript draft. AV and CB thank Florian Maumus for advice regarding TE annotation. The primary funding for this work was provided by KAUST award FIC/2010/09 to JL, MO, and AP.

## Additional information

### Funding

| Funder | Grant reference | Author |
| --- | --- | --- |
| King Abdullah University of Science and Technology (KAUST) | FIC/2010/09 | Aswini K Panigrahi, Julius Lukeš, Miroslav Oborník, Arnab Pain |
| Council of Scientific and Industrial Research | BSC0124 | Dhanasekaran Shanmugam |
| National Institute of Allergy and Infectious Diseases (NIAID) | HHSN272200900018C | Daniel E Neafsey |
| Australian Research Council (ARC) | DP120100599 | Ross F Waller |
| Monash University | | Christian Doerig |
| National Health and Medical Research Council (NHMRC) | | Christian Doerig |
| Czech Science Foundation (Grantová agentura České republiky) | P506/12/1522, 13-33039S, P501/12/G055 | Jan Michálek, Jaromír Cihlář, Aleš Tomčala, Julius Lukeš, Miroslav Oborník |

The funders had no role in study design, data collection and interpretation, or the decision to submit the work for publication.

### Author contributions

YHW, Performed gene annotations; environmental perturbation and transcriptome profiling; invasion pathway and apical complex analysis; DNA- and RNA-binding protein analysis; cross-species transcriptome analysis, Coordinated the genome and transcriptome analyses, Wrote the initial manuscript, Wrote the final manuscript; HA, Performed gene annotations; invasion pathway and apical complex analysis; subcellular targeting prediction; curation of extracellular and cytoskeletal genes in apicomplexans; TDO, Performed genome assembly and gene prediction; gene annotations; gene family analysis; CMK, Performed endomembrane trafficking system analysis; MK, JC, Performed genome analysis; JM, Performed fatty acid biosynthesis; AS, Performed generation and maintenance of specimen, DNA and RNA extractions, library preparation and sequencing; validation of predicted genes; manual curation for gene predictions; DS, Performed global metabolic analysis, Commented and edited on versions of the draft manuscript; AT, Performed generation and maintenance of specimen, DNA and RNA extractions, library preparation and sequencing; manual curation for gene predictions; environmental perturbation and transcriptome profiling; AV, Performed transposable element analysis; SA, EH, JJ, MN, Performed generation and maintenance of specimen, DNA and RNA extractions, library preparation and sequencing; AB, Performed gene annotation validations; JC, Performed generation and maintenance

of specimen, DNA and RNA extractions, library preparation and sequencing; heme pathway and phylogenetic analysis; PF, Performed analysis of chromerid metabolism; commented and edited on versions of the draft manuscript; SGG, performed generation and maintenance of specimen, DNA and RNA extractions, library preparation and sequencing; commented and edited on versions of the draft manuscript; AH, Performed urea pathway and phylogenetic analyses; NJK, NS, Performed validation of RAP's localization in mitochondria; FDM, DM-S, NDR, JW, Performed comparative genome analysis; TM, Performed gene structure analysis; RN, Performed genome validation, annotation, and submission; AKP, Conceived the project; validation of predicted genes; EP-R, AR, Performed manual curation for gene predictions; AT, Performed MS and gas chromatography on fatty acid synthesis; DEN, Performed generation and maintenance of specimen, DNA and RNA extractions, library preparation and sequencing; contributed some raw sequencing reads data; CD, Performed comparative genome analysis, Commented and edited on versions of the draft manuscript; CB, Performed transposable element analysis, Commented and edited on versions of the draft manuscript; PJK, Performed genome analysis, Commented and edited on versions of the draft manuscript; DSR, Global metabolic analysis, Commented and edited on versions of the draft manuscript; JBD, Performed endomembrane trafficking system analysis, Wrote the initial manuscript, Commented and edited on versions of the draft manuscript; TJT, Performed extracellular protein analysis, Wrote the initial manuscript, Commented and edited on versions of the draft manuscript; RFW, Performed validation of RAP's localization in mitochondria, Wrote the initial manuscript, Commented and edited on versions of the draft manuscript; JL, Conceived the project, Commented and edited on versions of the draft manuscript; MO, Conceived the project, Analysis of chromerid metabolism, Commented and edited on versions of the draft manuscript; AP, Conceived the project, Wrote the initial manuscript, Commented and edited on versions of the draft manuscript, Co-ordinated the project

### Author ORCIDs

Yong H Woo, http://orcid.org/0000-0002-0338-6493
Javier del Campo, http://orcid.org/0000-0002-5292-1421
Arnab Pain, http://orcid.org/0000-0002-1755-2819

## Additional files

**Supplementary files**

• Supplementary file 1. Summary of the genome assembly and the annotated genes of *Chromera velia*, *Vitrella brassicaformis*. Details of transposable elements on the genome are shown in *Supplementary file 2*.

• Supplementary file 2. Summary of transposable elements on the *Chromera velia* and *Vitrella brassicaformis* genomes.

• Supplementary file 3. Genes encoding proteins involved in forming photosystems in *Chromera velia* and *Vitrella brassicaformis*.

• Supplementary file 4. Genes encoding enzymes involved in heme biosynthesis in chromerids.

• Supplementary file 5. Domains of extracellular proteins and example genes in chromerids. (a) Species abbreviations: *Perkinsus marinus*, *P. mar*; *Chromera velia*, *C. vel*; *Vitrella brassicaformis*, *V. bra*; and *Cryptosporidium parvum*, *C. par*. (b) Domain accession identifiers. Domain information can be retrieved at the NCBI Conserved Domain website: (http://www.ncbi.nlm.nih.gov/cdd). (c) At the time of publication this accession identifier was valid but the relevant entry could not be retrieved via the NCBI Conserved Domain website: (http://www.ncbi.nlm.nih.gov/cdd). (d) A domain having two cysteines and thus far found only as tandem arrays in proteins of *Chromera velia* (for example, Cvel_967). (e) Cysteine-rich domain found in *Cryptosporidium* oocyst wall proteins (COWP) and in coccidians. (f) Archaeal protease-type repeats first described in the *Cryptosporidium* predicted EC protein, cgd7_4560. The domain was previously described as 'A small domain with characteristically spaced cysteine residues that is fused to a papain-like protease domain in the secreted protein

AF1946 from *Archaeoglobus fulgidus* (*Templeton et al., 2004a*)'. (g) The domain was previously described as 'Domain typically with 6 cysteines, seen thus far mainly in animals with a few occurrences in plants. It is found in the sea anemone toxin metridin and fused to animal metal proteases, plant prolyl hydroxylases and is vastly expanded in the genome of *Caenorhabditis elegans* (*Templeton et al., 2004a*)'. (h) The domain was previously described as 'β-strand rich domain, predicted to form a β-sandwich structure that is found in bacterial secreted levanases and glucosidases (*Templeton et al., 2004a*)'.

• Supplementary file 6. Mitochondrial localization predictions of selected RAP genes. Various algorithmic methods were used to identify candidates for experimental validations in *Toxoplasma*. Classifications are given in the column 'Loc' as M-mitochondria; S- secreted; O-others.

## Major datasets

The following datasets were generated:

| Author(s) | Year | Dataset title | Dataset ID and/or URL | Database, license, and accessibility information |
|---|---|---|---|---|
| Yong H Woo *et al*, | 2015 | Chromera velia DNA and RNA sequencing reads | http://www.ebi.ac.uk/ena/data/view/ERP006228 | Publicly available at the EBI European Nucleotide Archive (Accession no: ERP006228). |
| Yong H Woo *et al*, | 2015 | Vitrella brassicaformis DNA and RNA sequencing reads | http://www.ebi.ac.uk/ena/data/view/ERP006229 | Publicly available at the EBI European Nucleotide Archive (Accession no: ERP006229). |
| Yong H Woo *et al*, | 2015 | Chromera Velia Genome Assembly | http://www.ebi.ac.uk/ena/data/view/CDMZ00000000 | Publicly available at the EBI European Nucleotide Archive (Accession no: CDMZ00000000). |
| Yong H Woo *et al*, | 2015 | Vitrella brassicaformis Genome Assembly | http://www.ebi.ac.uk/ena/data/view/CDMY01000000 | Publicly available at the EBI European Nucleotide Archive (Accession no: CDMY01000000). |

The following previously published dataset was used:

| Author(s) | Year | Dataset title | Dataset ID and/or URL | Database, license, and accessibility information |
|---|---|---|---|---|
| Hu G, Cabrera A, Kono M, Mok S, Chaal BK, Haase S, Engelberg K, Cheemadan S, Spielmann T, Preiser PR, Gilberger TW, Bozdech Z | 2010 | Perturbation Transcriptome of Plasmodium falciparum | http://www.nature.com/nbt/journal/v28/n1/extref/nbt.1597-S2.xls | Publicly available as a part of published dataset. |

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

**Appendix 1**

## Genome characteristics.

The statistics of the genome assembly and annotation are shown in *Supplementary file 1*. There was bacterial contamination in 20% and 80% of the sequence reads in *Chromera* and *Vitrella*, respectively. There was a high amount of low-complexity DNA sequence repeats and TEs in *Chromera* (*Supplementary file 1*). By various bioinformatics methods ('Materials and methods'), we generated assemblies containing 5953 and 1064 scaffolds for *Chromera* and *Vitrella*, respectively. The total number of predicted genes differed between *Chromera* and *Vitrella* primarily due to significant differences in TE gene content between the two chromerids but the number of expressed genes was similar (*Supplementary file 1*).

We examined how genomes of the chromerids and other species were organized (*Supplementary file 1*). The median gene length is roughly the same between the two chromerids. The number of introns in a given gene was similar between the chromerids, although the size of introns was larger in *Chromera* than in *Vitrella* (*Supplementary file 1*). Compared to these chromerids, the number of introns in Apicomplexa was drastically less, raising the possibility that introns were compacted and reduced during apicomplexan evolution, which would need to be confirmed with further detailed investigation. For many genes (13,912 and 17,569 respectively for *Chromera* and *Vitrella*), we were able to assign 5′ and 3′ UTRs, using strand-specific transcriptome (RNA-seq) data sets. The distance between the protein-coding genes in *Vitrella* was short (median 92 base-pairs (bp)), indicating compactness of its genome. On the other hand, such distance was longer in *Chromera* (median 989 bp). Determining whether the common ancestor of chromerids had a compact genome or not would require analysis of genomes from more closely related species. There are three possible orientations by closely spaced neighboring genes can be clustered, that is, those with short intergenic spaces between the gene boundaries: tandem, head-to-head, or tail-to-tail. In both *Chromera* and *Vitrella* genomes, closely spaced (<1000 bp) genes were in head-to-head orientation more often than expected by chance (data not shown). It was previously shown that many neighboring genes in head-to-head clusters showed correlated expressions across various conditions; however, most of the co-expressions were modest; instead, head-to-head clustering is a major mechanism for stabilizing transcription of genes in fundamental cellular processes rather than for co-regulating the two genes (*Woo and Li, 2011*; *Russell et al., 2013*). Head-to-head clustering probably provided evolutionary and regulatory stability to genes involved in fundamental cellular processes. Other related species had different gene orientations, for example, the dinoflagellate *Symbiodinium microtinum* has tandem clusters driven by tandem gene duplication (*Shoguchi et al., 2013*). Given the dynamic nature of genome organization, we propose that different groups of species evolved different strategies for genome organization (*Woo and Li, 2011*).

Repetitive sequences constitute a significant proportion of eukaryotic genomes (*Fedoroff, 2012*). Thus, they play a significant role in evolution of host genomes. Systematic TE clustering, classification, and annotation were performed on 1064 *Vitrella* scaffolds (72.7 Mb genome—72,700,666 bp) and 5953 *Chromera* scaffolds (193.6 Mb genome—193,664,168 bp) Chromera. In both species, Class I elements (*Tempel, 2012*) make up a larger proportion of the genome than Class II elements (*Tempel, 2012*) (*Supplementary file 2*). The RT domain variation shows that *Eimeria tenella* TEs grouped separately and are not related to chromerid TEs (*Supplementary file 2*), suggesting gains of TEs in *E. tenella* (*Reid et al., 2014*) independently from chromerids. *Vitrella* forms a separate clade in the phylogenetic analysis of the RT domains.

## Appendix 2

### Metabolism.

#### Materials and methods

#### Reconstructing global metabolic map based on KEGG

Global metabolic pathways were mapped to KEGG metabolic pathways for the predicted protein-coding gene sets for the 26 species. KEGG ortholog (KO) assignments for the respective proteomes were made using the KO identification tools available on the KEGG website (http://genome.jp/tools/kaas) (*Moriya et al., 2007*; *Kanehisa et al., 2014*), and then the assigned KO numbers were used to identify and map metabolic pathways using the 'Search and color pathway' tool available on the KEGG site (http://genome.jp/kegg/tool/map_path-way2.html). The output of this mapping exercise was then manually inspected to compile the set of enzymes present in all major metabolic pathways (*Figure 2—source data 1*).

As KEGG is very strict in mapping orthologs and assigning KO numbers (so as to minimize false positives), we found numerous pathway holes (missing enzymes), many of which were readily apparent as false negatives. In order to resolve this, we then resorted to identifying orthologs from OrthoMCL-DB for all the genomes compared here (*Chen et al., 2006*). The resulting ortholog assignments were then used to manually verify presence/absence of missing enzymes for filling pathway holes where possible. This curated data, based on both KEGG and OrthoMCL assignments, were used to generate the final mapping of enzymes to pathways, and using this info a metabolic pathway network was drawn to represent all major pathways involved in carbohydrate, energy, fatty acid, lipid, isoprene, steroid, amino acid, nucleotide, cofactor, polyamine, and redox metabolism (*Figure 2—figure supplement 3*).

Based on the enzymes mapped, we calculated the completeness of metabolic pathways by comparing the fraction of enzymes present for each pathway in each species. The complete set of enzymes mapped to each pathway (originally taken from KEGG and further curated to eliminate non-specific entries) is given in column B of *Supplementary file 3*. The fractional values were then color-coded and the resulting data are shown in *Figure 2B*. In order to visualize the retention, loss or gain of higher level metabolic functions, the fraction of enzymes mapped to these pathways is indicated as a pie chart for hypothetical ancestors of selected apicomplexan groups and chromerids (*Figure 2B*). We used presence of enzymes across the species and the phylogenetic relationship to infer presence of enzymes in the hypothetical ancestors based on Dollo parsimony (*Csuros, 2010*). Dollo parsimony is based on an assumption that it is unlikely that the same enzymes were gained multiple times independently in different lineages.

#### Phylogeny of heme pathway enzymes, the urea pathway CPS and enzymes involved in fatty acid biosynthesis.

Predicted proteins from *Vitrella* (Chromera heme pathway is already published [*Kořený and Oborník, 2011*; *Kořený et al., 2011*]) were searched for enzymes involved in the synthesis of tetrapyrroles (aminolevulunuic acid [ALA] synthase, ALAS; ALA dehydratase, ALAD; Porpho-bilinogen deaminase, PBGD; Uroporphyrinogen synthase UROS; Uroporphyrinogen decar-boxylase, UROD; Coproporphyrinogen oxidase, CPOX; Protoporphyrinogen oxidase, PPOX; and Ferrochelatase FeCH). All genes identified were aligned to the homologs available in public databases such as NCBI and JGI using Muscle (Edgar, 2004), with the alignments further edited in SeaView (*Gouy et al., 2010*). The results from these analyses are shown in *Supplementary file 4*. The same procedure but searching in the predicted proteomes of both chromerids was used to construct the alignment of carbamoyl phosphate synthases (CPS). Genes coding for enzymes containing ketoacyl synthase domain were searched using BLAST. Functional domains were searched using InterProScan (*Zdobnov and Apweiler, 2001*). Phylogenetic trees of all investigated enzymes were constructed using the ML approach

(RAxML [*Stamatakis, 2014*]), Bayesian inference (PHYLOBAYES [*Lartillot and Philippe, 2004*]), and a method designed to deal with amino acid saturation (AsaturA [*Van de Peer et al., 2002*]). ML trees were computed under the gamma corrected LG4X model of evolution as implemented in RAxML 7.4.8a using the rapid-bootstrap optimization algorithm in 1000 replicates. Bayesian phylogeny was inferred using empirical site-heterogenous model C40 as implemented in Phylobayes 3.2f. Two independent chains were run until they converged (i.e., maximum observed discrepancy was lower than 0.2), and the effective number of model parameters was at least 100 after the first 1/5 generation was omitted from topology and posterior probability inference.

AsaturA trees were computed using a Poisson corrected LG model and the support was assessed from 1000 replicates. Sequences from *Vitrella* (all enzymes under investigation) and *Chromera* (CPS and FAS enzymes) were inspected for the presence of N-terminal leader sequences using SignalP (*Bendtsen et al., 2004*) and TargetP (*Emanuelsson et al., 2007*) software respectively, suggesting targeting to either mitochondrion (with mitochondrial transit peptide) or plastid (with bipartite leader composed of ER signal peptide and transit peptide).

## Fatty acid synthesis pathway

*C. velia* cells were grown in the f2 medium. Cultures were kept in 25 cm² flasks under artificial light with photoperiod 12/12, light exposure between 70 and 120 μmol/m²/s and temperature of 26°C. 1 ml of *C. velia* stationary culture was added to each flask with 20 ml of f2 solution. The cultures were grown for one month to reach a high density of cells. Since triclosan is not soluble in water, dimethyl sulfoxid (DMSO) was used as a soluble mediator. Four experimental groups were established: control, control with DMSO, *Chromera* treated with triclosan in concentrations of 1 mM and 0.5 mM, respectively. After 16 days of incubation, cultures were harvested via centrifugation, and pellets were stored in −20°C for subsequent lipid extraction. Homogenization of algal sample was achieved by Mini-beadbeater (Biospec Products). Homogenates were dried and weighted. Lipids were extracted using on chloroform and methanol, as described before (*Folch et al., 1957*). An aliquot of 100 μl volume was subjected to HPLC ESI/MS. The technique was performed on an ion trap LTQ mass spectrometer coupled to Allegro ternary HPLC system equipped with Accela autosampler with the thermostat chamber (all by Thermo, San Jose, CA, USA). 5 μl of sample was injected into a Gemini column 250 × 2 mm i.d. 3 μm (Phenomenex, Torrance, CA, USA). The mobile phase consisted of (A) 5 mmol/l ammonium acetate in methanol, (B) water, and (C) 2-propanol. The analysis was completed within 80 min with a flow rate of 200 μl/min by following gradient of 92% A and 8% B in 0–5 min, then 100% A till the 12th minute, subsequently increasing the phase C to 60% till 50 min and holding for 15 min and then in at the 65th minute returning back to the 92:8% A:B mixture and 10 min to column conditioning. The column temperature was maintained at 30°C. The mass spectrometer was operated in the positive and negative ion detection modes at +4 kV and −4 kV with capillary temperature at 220°C. Nitrogen was employed as shielding and auxiliary gas for both polarities. Mass range of 140–1400 Da was scanned every 0.5 s to obtain the full scan ESI mass spectra of lipids. For investigation of the lipid molecules structures the collisionally induced decomposition multi-stage ion trap tandem mass spectra MS² in both polarity settings were simultaneously recorded with a 3 Da isolation window. Maximum ion injection time was 100 ms, and normalized collision energy was 35%. Major phospholipids, galactolipids, and neutral lipids molecular species that are detected were separated by reversed-phase HPLC. The structure of each entity was identified by MS² experiments in positive or negative mode. Peak areas for each detected lipid component were summarized and their relative contents estimated to sum of all obtained peaks.

Raw extracted lipids have to be transformed to methylesters of fatty acids (FAMEs) to enable application of the GC technique. For this purpose sodium methoxide was employed as a transesterification reagent, as previously described (*Zahradnickova et al., 2014*). FAMEs were then analyzed by GC/FID. Hydrocarbon with 26-carbon chain was chosen as an internal standard. The chromatography was performed using gas chromatograph GC-2014 (Shimadzu) equipped by with column BPX70 (SGE)—0.22 mm ID; 0.25 μm film; 30 m length. μl of derivatized

sample was injected via autosampler and injector AOC—20i (Shimadzu) to the column in split mode (split ratio 10). The temperature of the injector was 220°C. The starting temperature of the column was 120°C holding for 4 min. Then, the temperature increased to the 180°C in at the rate of 10°C per minute, and after that 7°C per minute to 230°C. Temperature of the flame ionization detector was 260°C. The whole analysis takes took approximately 20 min. $H_2$ was used as a carrier gas. For the identification of particular FAs, a mixture of 37 standards purchased from Supelco Inc. was used.

# Results and discussion

## Global metabolic map

Metabolic annotations based on ortholog assignments with KEGG and OrthoMCL database showed that chromerids contain all major primary metabolic pathways typically found in free-living unicellular eukaryotes (*Figure 2—figure supplement 3*). 2918 *Chromera* proteins and 2985 *Vitrella* proteins were assigned KO numbers, from which 432 *Chromera* (1.3% of proteome) and 425 *Vitrella* proteins (1.8% of proteome), respectively, were identified as enzymes with a metabolic function based on EC number association.

In support of their autotrophic lifestyle, the chromerids appear to be capable of generating de novo all primary carbon metabolites such as the various sugars and other reduced carbon compounds (presumably via photosynthesis and associated carbon fixation pathways), amino acids, nucleotides, fatty acids and lipids, isoprene and steroid derivatives, and most vitamins (except biotin and vitamin B12). These organisms are also capable of assimilating both nitrate and sulfite and can generate energy from photosynthesis as well as mitochondrial respiration. A full complement of enzymes involved in sugar and sugar derivative metabolism, such as glycolysis, Kreb's cycle, pentose phosphate pathway, inositol mono- and poly-phosphate formation, polysaccharide formation, and amino- and nucleotide-sugar formation, is encoded by the chromerids. Chromerids are also capable of synthesizing sulfoquinovosyl-diacyl-glycerol lipids, which are found associated with the chloroplast in photosynthetic organisms. *Figure 2—figure supplement 3* illustrates a complete representation of all major pathways mapped to chromerids in comparison to selected apicomplexan lineages.

Generally, *Chromera* and *Vitrella* have similar sets of metabolic enzymes. Enzymes for the oxidative arm of pentose phosphate pathway, conversion of diacyl-glycerol to phosphatidyl ethanolamine, phosphatidyl ethanolamine to phosphatidyl serine, and XppppX to XTP are absent in *Chromera*, while, on the other hand, the enzymes involved in conversion of glucose-1P to UDP-glucose, and cytidine to uridine are missing in *Vitrella*. One major difference between the two chromerids is that the complex III of the mitochondrial respiratory chain (cytochrome *c* reductase) is missing in *Chromera*, but present in *Vitrella* (*Flegontov et al., 2015*). This feature of the *Chromera* mitochondrion, that is, absence of complex III but presence of complex IV, makes it unique among all mitochondria and mitochondria-derived organelles.

The crucial enzyme for the urea pathway, mitochondrially targeted carbamoyl phosphate synthase (CPS) (*Allen et al., 2011*), is absent from both chromerids. However, while *Chromera* contains single CPS involved in pyrimidine biosynthesis, *Vitrella* genome encodes two CPSs. But both these genes are closely related suggesting they are recent duplicates (ML tree is shown in *Figure 2—figure supplement 2A*) and they lack a mitochondrial leader sequence at the N-terminus (data not shown). This means that *Vitrella* duplicated CPSs are not likely to be involved in urea cycle. In contrast to *Vitrella*, *Chromera* lacks the gene encoding argininosuccinate lyase (ASL), an enzyme of the ornithin cycle.

## Plastid-related metabolic pathways

Chromerid photosystems have a reduced set of genes similarly to that of other algae with a complex plastid. The patterns of reduction were lineage-specific, even between the two known chromerids. We found *psbM*, *petN*, *psaJ*, *Psb27*, *Ycf4*, and *Ycf44* genes in *Vitrella* but absent in *Chromera*; vice versa, *Ycf39* and *Ycf54* are absent in *Vitrella* but present in *Chromera* (**Supplementary file 3**). This demonstrates that the plastid of *Chromera* is more diverse and reduced when compared to *Vitrella* with respect to both the composition of photosystems and the number of genes encoded in the plastid genome (**Janouškovec et al., 2010**). In spite of substantial reduction of photosystems, photosynthesis in *Chromera* is highly efficient (**Quigg et al., 2012**).

The heme pathway in *Vitrella* is homologous to that found in *Chromera* (**Kořený et al., 2011**) for most of the involved enzymes (ALAS, ALAD, PBGD, UROS, PPOX), however, *Vitrella* and *Chromera* do not constitute sister groups in the CPOXs and FeCH trees (trees not shown). Some enzymes are not present in single copies (UROD) in *Vitrella*, in contrast to *Chromera*, where three orthologs originating from cyanobacteria, endosymbiont nucleus, and exosymbiont nucleus are present (**Kořený et al., 2011**). For some investigated enzymes (UROS, UROD, CPOX), only the endoplasmic reticulum signal peptide was found with transit peptide missing from the sequence, suggesting their possible location in the endoplasmic reticulum or periplastidal space.

Genes containing the ketoacyl synthase domain and thus likely involved in fatty acid or polyketide synthesis were searched in the genomes of chromerids. We found that both algae possess multi-modular enzymes responsible for fatty acid synthesis type I (FASI), similar to some apicomplexan parasites, such as *Cryptosporidium* spp. and *T. gondii*. The longest multi-modular enzyme found in *Chromera* contains five multi-domain modules, reaching over 11,000 amino acids in length (**Figure 2—figure supplement 2B**).

## Evolution of metabolic pathways

Apicomplexan parasites differ drastically from each other in their metabolic functions, and have a significantly reduced metabolic capability in comparison to the chromerids. Apicomplexans are non-photosynthetic and therefore lack all associated metabolic activities including photosynthetic carbon fixation. Interestingly, however, all plastid-bearing parasites have retained only the ferredoxin-NADP+ reductase (FNR)/ferredoxin redox system of the photosynthetic electron transport (**Lim and McFadden, 2010**). In photosynthetic organisms, this redox system mediates the transfer of electrons originating from water to NADP+, resulting in the formation of NADPH (cofactor for fatty acid biosynthesis and Calvin cycle), and it is likely that this role is conserved in chromerids. In apicomplexans, the source of electrons for generating reduced ferredoxin is not clear, but it is evident that reduced ferredoxin is required for generating reducing equivalents and is a cofactor for several plastid-associated enzymes, including those involved in isoprene synthesis (**Lim and McFadden, 2010**). Other notable pathways missing in apicomplexans but present in the chromerids include the following: glyoxalate pathway; steroid biosynthesis; synthesis of aromatic and branched-chain amino acids; purine synthesis; and synthesis of cofactors such as thiamine, riboflavin, and nicotinate ribonucleotide.

Despite the reduced metabolic capabilities, certain core metabolic functions are conserved in chromerids as well as in all apicomplexan parasites, and many of these are likely to be essential. These pathways include: glycolysis; synthesis of ubiquinone, inositol-P derivatives, GPI-anchor, mono-, di- and tri-acyl glycerol, isoprene derivatives, and N-glycans; a subset of scavenge reactions for purine and pyrimidine bases and their conversion to nucleotides; glycine–serine inter-conversion; one-carbon folate cycle and S-adenosyl-methionine formation. There are many metabolic pathways that are retained in specific apicomplexan lineages but shared with the chromerids (see **Figure 2B** and **Figure 2—figure supplement 3**). The following are notable examples of pathways shared between chromerids and a apicomplexan lineage: with *Plasmodium*, polyamine synthesis; with *Cryptosporidium*, conversion of serine to tryptophan; with *Toxoplasma*, branched-chain amino acid degradation, synthesis of aspartate, lysine, and methionine; synthesis of molybdopterin, biopterin, pyridoxal-phosphate, and pantothenate

cofactors. Surprisingly, with respect to the chromerids and other apicomplexans, *Cryptosporidium* appears to be the only apicomplexan lineage to have gained a metabolic function of conversion of thymidine to dTMP by thymidine kinase. In addition, the type I and II pathways for fatty acids biosynthesis show lineage-specific distribution in apicomplexans and chromerids (*Figure 2—figure supplement 3*).

We can also find example of pathways that have been lost in a lineage-specific manner. For example, the ability to synthesize the di-saccharide trehalose is missing only in *Plasmodium*. However, the most dramatic loss of metabolic function in a single lineage can be found in cryptosporidia. These parasites are devoid of all plastid- and mitochondria-associated metabolic functions and other pathways involved in the synthesis of ribose-P, pyrimidine, most amino acids, heme, fatty acids (de novo), and isoprene units. It seems that the lack of mitochondrial oxidative pathways in cryptosporidia led to loss of the ability to generate flavin nucleotide (FMN/FAD) and lipoic acid cofactors.

In order to cope with loss of metabolic pathways, parasites have evolved various mechanisms for scavenging the required nutrients and metabolites from their respective hosts. For example, metabolites such as heme, fatty acids, steroids (specifically cholesterol), and sphingolipids are known to be scavenged by various apicomplexans as needed from their respective hosts.

According to the metabolic pathway maps, certain metabolic functions, which are coupled to each other, have been either retained or lost concomitantly in various species (*Figure 2—figure supplement 3*). For example, piroplasms and cryptosporidia lack de novo fatty acid biosynthesis along with the pyruvate dehydrogenase enzyme complex (plastid-associated in apicomplexans), which is known to supply acetyl CoA units for fatty acid synthesis. On the other hand, these parasites have retained the ability to convert pantothenate to co-enzyme A, which is required for the activation of fatty acids scavenged from the hosts (*Leonardi et al., 2005*). Similarly, activities of the serine hydroxyl methyl transferase and thymidylate synthase enzymes are coupled to each other and to one-carbon folate metabolism. Therefore, these three metabolic functions are retained in all parasites.

## Appendix 3

# Endomembrane trafficking system.

### Materials and methods

The predicted proteomes of the 26 species in **Figure 1** have been searched for endomembrane trafficking components. Initial homology searching was carried out using BLAST (**Altschul et al., 1990**). Known sequences from human (*Homo sapiens*) and yeast (*Saccharomyces cerevisiae*) were used to search the proteomes of each organism including *Chromera* and *Vitrella* to identify potential homologs of proteins implicated in endomembrane trafficking. Any sequences scoring an initial E value of 0.05 or lower were subjected to confirmation by reciprocal BLASTP. This involved the use of candidate homologous sequences as queries against the relevant *H. sapiens* or *S. cerevisiae* genome. Sequences that retrieved the query sequence, or named homologs/paralogs/isoforms thereof, first with an E value of 0.05 or lower were considered true homologs.

Additional searches were carried out using HMMER (**Finn et al., 2011**). The HMMs for the initial queries were built and used to search each proteome. Top hits based on BLASTP results with E values less than 0.05 were considered confirmed homologs, and not subjected to further analysis. Subunits with significant HMMER hits were further investigated by the reciprocal BLASTP as described above. Further HMMER searches were carried out with the addition of homologous sequences from *Bigelowiella natans*, *Phytophthora infestans*, and *T. gondii* to the original HMMs. Results were analyzed identically to the first round. All identified endomembrane components are listed in **Figure 2—source data 2**.

To identify homologous proteins not predicted by the gene prediction software, we used TBLASTN with the homologous protein from the closest related organism in our data set against scaffolds and contigs; E value cut-off was identical to BLASTP analysis. We utilized BLASTP to search either genome with an identified homolog from the other, if it was present. The final results are summarized in **Figure 2—figure supplement 4** using the Coulson Plot Generator software (**Field et al., 2013**).

### Results and discussion

Apicomplexa possess numerous unusual features in their membrane trafficking systems. Non-canonical membranous inclusions such as the invasion organelles, the micronemes, rhoptries, and dense granules are present (**Baum et al., 2008**). Though canonical, stacked, Golgi bodies are present in *T. gondii* (**Pelletier et al., 2002**), other apicomplexan species possess Golgi bodies with aberrant morphology and unusual characteristics (**Struck et al., 2008**). Combined with other organelle destinations such as mitochondria, digestive vacuoles involved in hemoglobin catabolism in *P. falciparum*, and plant-like lytic vacuoles in *T. gondii* (**Miranda et al., 2010**), specificity of protein and lipid components of these various organelles suggest a need for unique trafficking pathways mediated by distinct protein machinery.

Interestingly, previous studies demonstrated the loss of trafficking machinery in Apicomplexa, including three key sets of proteins in the ESCRT machinery (**Leung et al., 2008**), adaptor protein complex (AP) families (**Nevin and Dacks, 2009**; **Hirst et al., 2011**), and multi-subunit tethering complexes (MTCs) (**Koumandou et al., 2007**; **Klinger et al., 2013a**) have been published. Several of the aforementioned families are involved in trafficking within the late endosomal system in opisthokont models and so may be associated with the evolution of the rhoptries and micronemes within the apicomplexan or myzozoan lineage. Consistent with this

idea, some cases of reduction were not limited to Apicomplexa, and could be observed in the sister phyla of the ciliates and dinoflagellates.

This pattern of loss raises the question of what losses correlate with the transition to parasitism and which are pre-adaptive, arising more deeply in the lineage. The unique phylogenetic position of chromerids (*Janouškovec et al., 2010*, *2012*; *Oborník et al., 2012*) allows finer dissection of the patterns of retention/loss observed previously. Hence, we chose to focus on detailed characterization of the three previously studied sets of membrane trafficking machinery in the predicted proteomes of *Chromera* and *Vitrella*, together with 24 closely related organisms for comparison.

## ESCRT machinery

The ESCRT machinery is a set of five sub-complexes involved in recognition of ubiquitylated proteins and recruitment to the multi-vesicular body (MVB)/late endosome for degradation (*Leung et al., 2008*). Most eukaryotes, including *Chlamydomonas reinhardtii* and our representative stramenopile taxa (*Thalassiosira pseudonanna*, *Phaeodactylum tricornutum*, *Ectocarpus siliculosus*, and *Pythium ultimum*), have a complete set of the ESCRT machinery, suggesting that the ancestor of alveolates, and indeed the Last Eukaryotic Common Ancestor (LECA), likely had it. Though this ancestral complement appears to have been reduced in ciliates in the ESCRTI and III complexes, and a few components are missing from dinoflagellate taxa, numerous gene duplications have occurred as well, suggesting sculpting of the machinery. By comparison, apicomplexan parasites exhibit significant reductions in their ESCRT machinery (*Leung et al., 2008*). Cryptosporidia, coccidia, and plasmodia appear to lack any subunits of the ESCRTI and II complexes. ESCRTIII conservation is better, though no apicomplexan encodes Vps24, and multiple taxa have lost Vps20 as well. A similar pattern is seen for the ESCRTIII-a machinery, with piroplasmids encoding only Vps46 and Vps4. Coccidia additionally encode Vps31, and cryptosporidia Vps60, whereas plasmodia encode all subunits (rodent parasites like *Plasmodium chabaudi*), or lack Vps31 (human or simian parasites like *P. falciparum*). *Chromera* and *Vitrella* possess all ESCRT subunits except for the ESCRT-III component CHMP7, which is rarely found outside the opisthokont supergroup (*Leung et al., 2008*). This observation suggests two conclusions regarding the evolution of the ESCRT machinery within alveolates: massive gene loss within the Apicomplexa occurred recently, after the split from the proto-apicomplexan ancestor, and some losses of machinery shared between apicomplexans and other alveolates are due to independent losses. An excellent example of this latter case is that of Vps37, which is present only in chromerids, but in no other alveolate included in the current study, suggesting its function was dispensable in a large number of lineages.

## APs

The APs are heterotetrameric complexes that select cargo for inclusion into transport vesicles at organelles of the late secretory system and endocytic system. AP1 and AP3 are involved in the transport between the trans-Golgi network (TGN) and endosomes. AP2 is involved in the transport from the cell surface. AP4 is involved in TGN transport to either endosomes or the cell surface, while the recently described AP5 complex is involved in the transport from late endosomes back to early endosomes. All five complexes are ancient, having likely been present in the LECA (*Nevin and Dacks, 2009*; *Hirst et al., 2011*). However, the complexes have also been secondarily lost on multiple occasions as well. Outgroup taxa in our data set possess AP1-4 complexes, with the exception of *C. reinhardtii* lacking AP3, but only *Symbiodinium minutum* possesses an AP5 complex.

Apicomplexa display higher variability in AP complex retention. With the exception of AP2M in cryptosporidia, all taxa retain full AP1, 2, and 4 complexes. Piroplasms lack all subunits of the AP3 complex, and together with *P. falciparum* and *Plasmodium reichenowi*, lack AP5 as well. Other plasmodia possess all AP5 subunits with the exception of the mu subunit. This result was unexpected, based on the usual patterns of conservation seen across *Plasmodium* species.

Presence of AP5 in the majority of these organisms suggests the exciting possibility of a novel trafficking pathway absent from the comparatively well-studied human parasite *P. falciparum*. Additionally, our increased taxon sampling has suggested that AP5 may be well conserved across Myzozoa, a result otherwise indeterminable from previous studies of this protein family (**Hirst et al., 2011**). Cryptosporidia also lack AP3, but unlike piroplasmids, they possess almost a complete AP5 complex, missing only the sigma subunit. Coccidia are the exception, possessing all five AP complexes in their entirety. Excitingly, *Chromera* and *Vitrella*, like coccidia, possess a complete complement of adaptin subunits, suggestive of a more complete set of trafficking pathways to endosomal organelles in these organisms.

## MTCs

The MTCs are an assembly of heteromeric protein complexes involved in the first stage of vesicle fusion and delivery of contents from a transport vesicle to a destination organelle. Each one is specific to an organelle or transport pathway and all eight complexes have been deduced as present in the LECA, with some interesting cases of secondary loss. While *C. reinhardtii* and the stramenopiles encode a complete set of MTC machinery, several of these MTCs have interesting patterns of conservation, specifically in the Apicomplexa (**Klinger et al., 2013a**).

The conservation of the TRAPP I–II complexes is unclear through eukaryotes and clear patterns are difficult to draw. However, the apparent absence of the entire TRAPPII complex in *Vitrella* may be due to gaps/biases/absences in sequencing, protein prediction, or analysis, but has interesting ramifications if proven to be a real biological phenomenon.

Exocyst is involved in diverse processes, all of which involve polarized exocytosis (**Liu and Guo, 2012**). *Tetrahymena* appears to encode only four of the Exocyst subunits. None of the eight subunits were identifiable in *Chromera*, *Vitrella*, nor in any of the Apicomplexa or dinoflagellates. This confirms, and extends, a previous result suggesting the absence of this complex within the Myzozoa, suggesting a bona fide ancestral loss concurrent with the acquisition of an apical complex that could have served an analogous tethering function for secretory organelles.

COG is an octameric complex involved in tethering at the Golgi body (**Tomavo et al., 2013**; **Klinger et al., 2013b**). The COG complex is poorly conserved in Apicomplexa and a ciliate *Tetrahymena thermophila* only encodes half of the COG subunits. In contrast, all eight COG subunits are present in *Chromera* and *Vitrella*. The retention of a complete COG complex in both *Chromera* and *Vitrella* contrasts with the substantial loss of subunits in Apicomplexa, especially outside the coccidians (**Klinger et al., 2013b**) (**Figure 2—figure supplement 4**). Notably, this conservation is consistent with the presence of robust, stacked Golgi bodies in *Chromera* (**Oborník et al., 2011**) and *T. gondii* (**Pelletier et al., 2002**), compared to aberrant morphology in other Apicomplexa.

The complexes of CORVET and HOPS mediate tethering at the early and late endosomes (**Tomavo et al., 2013**; **Klinger et al., 2013b**). They share a core of four subunits with complex-specific proteins (Vps3 and 8 for CORVET and Vps39/41 for HOPS). Though all taxa encode the complete VpsC core of the HOPS/CORVET complex, all taxa except for *T. gondii* only appear to encode the CORVET-specific interactor Vps3. *Chromera* and *Vitrella*, like Apicomplexa, possess the entire VpsC core complements as well as the HOPS component Vps41 and both CORVET components.

Chromerids exhibit complex life cycles, from immotile vegetative cells to multi-cellular sporangia, and occasionally motile flagellated cells. Both lineages contain numerous potential locales for intracellular trafficking including mitochondria, plastid, starch granules, flagella, micronemes, and, in *Chromera*, the chromerosome. Additionally, vesicular traffic to the sporangial/cyst wall has been visualized in both lineages (**Oborník et al., 2012**). Our results indicate that chromerids possess an appropriately complex complement of membrane trafficking machinery to achieve these requirements.

Though MVBs have not been explicitly imaged or characterized in either lineage to date, both *Chromera* and *Vitrella* encode a complete set of ESCRT machinery, suggestive of the presence of functional MVBs. These may play a key role in modulating surface protein expression in various life cycle stages. Importantly, the close evolutionary position of *Chromera* and *Vitrella* to Apicomplexa suggests that the extensive decrease in ESCRT subunit conservation in Apicomplexa occurred in the immediate ancestor and is not an ancestral feature of a more inclusive group (**Leung et al., 2008**) (**Figure 2—figure supplement 4**). Particularly, the lack of some ESCRT subunits such as Vps37 in ciliates and dinoflagellates is most parsimoniously attributed to multiple independent losses. Further evidence for a complete set of ESCRT machinery in the last common alveolate ancestor comes from the conservation of all subunits to the exclusion of CHMP7 in the outgroup stramenopile taxa and in *C. reinhardtii*. The absence of CHMP7 in all taxa is not unusual, as it is lost in numerous taxa across eukaryotes (**Leung et al., 2008**).

Conservation of adaptin subunits is striking, particularly the complete retention of AP5 in chromerids. In an initial study of seven organisms from the SAR supergroup (the group in which chromerids belong to), only two (*B. natans* and *T. gondii*) were found to encode the complex; conservation across eukaryotes was similarly sparse (**Hirst et al., 2011**). The presence of a complete AP5 complex in chromerids and coccidians may be indicative of a conserved function in both lineages. Likewise, the retention of an almost complete AP5 in cryptosporidia and plasmodia may have functional significance or may simply represent a reductive evolutionary process that has not yet reached completion. The complete lack of AP5 in *P. falciparum* and *P. reichenowi* supports the latter view. As with the ESCRT complexes, the presence of AP1-5 in chromerids suggests the loss of AP3 and AP5 observed in some Apicomplexa is secondary, as well as the loss of AP5 in *Perkinsus marinus*, and in both ciliate lineages.

Presence of a complete VpsC core along with an additional CORVET subunit Vps3 in the majority of apicomplexan genomes suggests the potential for a modified HOPS/CORVET complex that interacts with Rab5 to direct tethering at the micronemes/rhoptries. This is in keeping with the view of rhoptries/micronemes as divergent endolysosomal organelles (**Klinger et al., 2013b**). However, chromerids do not appear to possess rhoptries, although chromerids possess cellular components analogous to micronemes (**Oborník et al., 2011**, **2012**). More HOPS/CORVET subunits were found to be conserved in *T. gondii*, which are the only apicomplexan to date to be described as possessing a canonical lysosome-like compartment[5], suggesting that complete complexes are retained in these lineages because they are required for trafficking to canonical lysosome-related organelles as well. Additionally, *Chromera* possesses the chromerosome, which often displays intralumenal vesicles similar to MVBs, suggesting it may also be derived from endosome-like organelles (**Oborník et al., 2011**).

In conclusion, apicomplexans possess unusual endomembrane compartments including atypical Golgi and endosome-derived invasion organelles such as micronemes and rhoptries (**Klinger et al., 2013b**). Modifications in the complement of membrane trafficking machinery, including the loss of key protein complexes found in most eukaryotes, have been observed in the apicomplexan lineage, potentially associated with the specialization of the endomembrane system. The absence of some components (Exocyst, Vps39, Trs120, Tip20) within *Chromera* and *Vitrella* suggests pre-adaptation to parasitism deeper in the apicomplexan lineage. By contrast, the presence of near complete complements of key machinery (AP1-5, ESCRTs, COG) absent in many apicomplexans, pinpoints the timing of the losses at the colpodellid/apicomplexan transition.

Appendix 4

## Apical complex and cytoskeleton.

Motility is an essential feature of many living organisms. Some organisms utilize microtubule-based specialized structures such as flagella and cilia for locomotion. Some use actin-based structures like filopodia, lamellipodia, and pseudopodia (*Frenal and Soldati-Favre, 2009*), which are exploited in the amoeboid crawling (*Pollard and Borisy, 2003*) or bacterial and viral movement into and between cells (*Stevens et al., 2006*). Apicomplexan parasites use an unconventional actin-based mode of locomotion known as gliding motility (*Morrissette and Sibley, 2002*). This mechanism allows the parasites to move fast in the absence of canonical microtubular and actin-based structures. Gliding motility is mediated by the apical complex, which is a cellular structure common to all apicomplexan parasites. In the apical complex, proteins secreted from specialized secretory organelles, microneme and rhoptries, mediate adhesion to the cell substrate during motility and invasion or formation of a PV (*Baum et al., 2006*).

Actin-based gliding motility is essential for apicomplexan invasion (*Skillman et al., 2011*). Apicomplexan gliding motility undergoes actin polymerization/depolymerization for their directional motility with other associated protein classes such as actin-like proteins (ALP), actin-related protein (ARP), capping protein (CP), formin, profilin and cofilin/ADF. Actins elongate in the form of filaments and push the membrane forward. Arp2/3 complex (one of the ARPs) mediates the initiation of new branches on pre-existing filaments. After some growth, CP terminates the elongation of the filaments. Cofilin/ADF promotes de-branching and de-polymerization. Profilin mediates the refilling of ATP-actin monomer pools, which are used for elongation through catalyzing ADP-ATP exchange (*Baum et al., 2006*; *Foth et al., 2006*).

We identified and compared genes encoding actins and other related components in the 26 species according to a method described by a previous study (*Baum et al., 2006*). Chromerids share homology with Apicomplexa for most of the actin, ALP and ARP classes. For example, both chromerids possess actin 1 (ACT1), actin-related (ARP), and actin-like (ALP) homologs. There were fewer actin genes in apicomplexans than in chromerids, indicating losses during apicomplexan evolution. The patterns of losses were the same for closely related species, suggesting non-random, lineage-specific losses (*Figure 2—figure supplement 5C*).

Arp2/3 complex, a nucleator of actins (*Machesky et al., 1994*), consists of seven subunits that regulate actin polymerization (*Mullins et al., 1997*; *Fehrenbacher et al., 2003*). Initially identified in *Acanthamoeba* (*Machesky et al., 1994*), Arp2/3 complex is conserved in most eukaryotes (*Gordon and Sibley, 2005*). We could not identify genes encoding subunits of Arp2/3 complex in both chromerids (*Figure 2—figure supplement 5C*). Also, all subunits were not found in apicomplexan species, consistent with a previous study (*Gordon and Sibley, 2005*). Individual subunits are important, as subunit ARPC4/p20 was shown to be essential for a complete, functional Arp2/3 complex (*Gournier et al., 2001*). Different subunits were identified in different phyla (*Figure 2—figure supplement 5C*). Within Apicomplexa, ARPC1 and ARPC4 were present in *Cryptosporidium hominis* and *Cryptosporidium parvum*, and ARPC1 and ARPC2 in *Plasmodium* spp. *S. minutum*, a dinoflagellate, contains genes for most of the subunits except ARPC1. This suggests that the common ancestor of Myzozoa (chromerids, apicomplexans, and dinoflagellates) had all the subunits, and they were lost in different lineages. Genes encoding ALP1, hypothesized to function as Arp2/3-like nucleator (*Gordon and Sibley, 2005*), were found in apicomplexans and also in *Vitrella* (Vbra_266.t1). FH2-domain (Pfam-PF02181) containing formins are members of another actin nucleator gene family. They produce unbranched filaments unlike Arp2/3 complex, which induces branched filaments. Both chromerids possess formin1 (FRM1) and formin 2 (FRM2) homologs, which are conserved in all the other studied species as well. Although *Plasmodium* spp. maintained a 1-1 orthology for both FRM1 and FRM2, we found a coccidian-specific FRM3 (TGME49_213370), suggesting a lineage-specific expansion. Maintenance of some formins across chromerids and

apicomplexans and lack of Arp2/3 complex suggest their importance, perhaps reflecting a switch from Arp2/3 complex to formins for actin nucleation during the evolution of Apicomplexa. Taken together, it seems that an Arp2/3 independent actin nucleation mechanism had already evolved before Apicomplexa and chromerids, and losses of ARP have probably begun too, although inferring the exact timing and sequence of losses will require studying more closely related species such as Colpodella.

We analyzed coronins, a major conserved gene family with a multifunctional role in actin regulation and vesicular transport (*Rybakin and Clemen, 2005*). These are WD40-repeat containing proteins, which represent the only candidate for actin bundling in apicomplexan parasites. Coronins inhibit the nucleating activity of Arp2/3, unlike other known Arp2/3-binding proteins. We observed absence of coronins in both chromerids, which is consistent with the notion that they, functionally linked to the Arp2/3 complex, were lost (*Figure 2—figure supplement 5C*). Although parasite homologs do not have the microtubule-binding domain of canonical coronins, but essential amino acid residues are conserved (*Gandhi et al., 2010*). Thus, coronin could be playing a role in stabilizing F-actin scaffolds or having an alternative role in vesicular transport in apicomplexan parasites.

Profilins are actin-binding proteins that supply pools of readily polymerizable actin monomers (*Baum et al., 2006*). Genes encoding profilins were found in all 26 species studied except for an oomycete (*P. ultimum*) and diatoms (*P. tricornutum*, *Thalassiosira pseudonana*). Apicomplexa-specific profilins have βmini1 and βmini2 domains, which provide an extended interface with actin and formed the structural basis of their actin-binding function in *Toxoplasma* (*Kucera et al., 2010*) and *Plasmodium* (*Kursula et al., 2008*). These domains are not found in other eukaryotes. Sequence alignment of these profilins reveals an intriguing observation that *Vitrella* (Vbra_7301.t1) had these β-domains previously thought to be specific to Apicomplexa, with partial conservation in Chromera (Cvel_18957.t1) and in dinoflagellate *P. marinus* (XP_002774080). The β-domains were not detected in other non-apicomplexan species. All species studied has had only one profilin gene except for chromerids where we observed 2 in *Chromera* and 3 in *Vitrella*, including an one-to-one ortholog of the apicomplexan profilin in both chromerids.

Cyclase-associated proteins (CAPs) are evolutionary conserved G-actin-binding proteins, which participate in filament turnover regulation by acting on actin monomers (*Chaudhry et al., 2010*). CAP proteins are made up of three significant regions: N-terminal adenylate cyclase binding domain (CAP_N, linked to the cAMP-RAS signaling), a central proline-rich segment, and a C-terminal actin-binding domain (CAP_C). Apicomplexans do not possess the N-terminal (CAP_N) domain altogether with few genes in stramenopiles and in *Vitrella* (Vbra_7026.t1) also showing a similar pattern of loss. However, the *Chromera* gene Cvel_8488.t1 possesses both domains. This suggests the dispensable nature of CAP_N domain (*Figure 2—figure supplement 5C*), and we speculate that in parasites CAP functions are reduced to actin sequestration only.

The F-actin capping, CapZ duplex, a dimer of α- and β-CPs, prevents polymerization from the 'barbed' (plus) end. It is conserved across Apicomplexa except for in piroplasms. It is also conserved in most of the species studied including stramenopiles, dinoflagellates, and both chromerids. In Apicomplexa, several gelsolin domain-containing proteins were found but they are unlikely to be functionally related and are speculated to be Sec23/Sec24-like proteins, which function in vesicular transport (*Baum et al., 2006*).

Cofilin/ADF genes promote de-branching of actin filaments and are well conserved among species studied. However, plasmodia differ from the rest of the Apicomplexa species in having an additional copy of the ADF gene. Phylogenetic analysis shows that ADF in plasmodia has duplicated and diverged with respect to the rest of the Apicomplexa, and recent structural studies explain the mechanism of action of *Plasmodium* ADFs (*Singh et al., 2011*; *Wong et al., 2014*). This represents a clear example of additional innovations of actin regulation in certain apicomplexan clades.

In addition, we identified myosin families in the 26 species using a myosin HMM model (*Foth et al., 2006*) (*Figure 2—figure supplement 5C*). Members of piroplasmids such as *Theileria annulata* and *Theileria parva* have the fewest genes among the apicomplexan species examined, likely because piroplasms do not require motility for intracellular invasion. On the other hand, we detected the most complete myosin family repertoire in *Chromera* and *Vitrella*. We detected certain myosin families in some apicomplexan genera, but not among non-apicomplexan species, indicating lineage-specific gains (data not shown). In summary, combinations of lineage-specific losses and gains have led to streamlined, unique repertoires of actins and myosins in various apicomplexan species.

## Appendix 5

# Extracellular proteins in chromerids.

We curated the chromerid genomes for genes with extracellular domains and domain architectures like similar to those of apicomplexans (*Figure 3—figure supplement 4*; *Supplementary file 5*). Both chromerids possess mucin-like proteins having long stretches of threonine and serine residues with predicted O-linked glycosylation, as well as the enzyme pathways involved in O-linked glycosylation (*Templeton et al., 2004a*; *Anantharaman et al., 2007*). Proteins encoding combinations of von Willibrand factor A (vWA) and thrombospondin 1 (TSP1) were also observed, although none with apparent orthologous relationship to the vWA and TSP1 domain proteins (TRAP) that serve as receptors mediating gliding motility in apicomplexans. The chromerid genomes possess numerous secreted proteins with domains predicted to participate in binding of sugar moieties (*Figure 3—figure supplement 4*). Chromerids share FRINGE domains with *Cryptosporidium*, and HINT domains with *Cryptosporidium* and *Gregarina*, to the exclusion of other apicomplexans, in support of early divergence of these genera within the Apicomplexa (*Figure 3—figure supplement 4B*). *Vitrella* genome contains multiple copies of proteins, which have arrays of the cysteine-rich oocyst wall protein (OWP) domain found in *Cryptosporidium* and coccidians, which are associated with forming environmentally durable walls of oocysts (*Templeton et al., 2004b*).

Several EC domain architectures thought to be distributed apicomplexan-wide have homologs in the chromerids; for example, the LCCL domain-containing proteins, CCp1 and CCp2/3, as well as the CPW-WPC domain proteins (*Figure 3—figure supplement 4C*). Ultrastructures reminiscent of micronemes have been observed in both chromerids (*Oborník et al., 2012*); consistent with this, we identified EC proteins having domains and architectures typical of *Toxoplasma* and *Plasmodium* micronemal secretory proteins. Examples include expansions of proteins containing SUSHI, EGF, TSP1, and vWA domains (data not shown). Chromerids possess unique architectures of proteins containing the macrophage perforin (MacPerf) domain (*Figure 3—figure supplement 4E*), which, previously found in apicomplexans and ciliates (as large expansions), are thought to function in apicomplexans to mediate membrane lysis during host cell egress and tissue traversal (*Roiko and Carruthers, 2009*). The *Chromera* MacPerf domain proteins also contain arrays of a domain, WSC, thus far not found in other alveolates, as well as a unique C-terminal DERM domain. *Chromera* possesses four MacPerf domain proteins with various domain architectures, whereas *Vitrella* a single MacPerf protein with a stand alone MacPerf domain (Vbra_18070.t1).

The ciliate genomes possess highly amplified and antigenically diverse repertoires of GPI-linked proteins termed 'immobilization antigens' (*Caron and Meyer, 1989*). We did not see amplifications of GPI-linked gene families in either chromerid species. Lineage-specific gene amplifications include a predicted secreted protein in *Vitrella*, which contains an arenylsulfonase domain (*Figure 3—figure supplement 4E*). The chromerids possess highly amplified gene family, annotated as 'CAST multi-domain protein' in the ciliate, *Oxytricha* (e.g., UniProt ID: OXYTRI_15408), and which comprises conserved cysteine-rich domains in the extracellular region, a single transmembrane domain, and a conserved predicted coiled-coil region in the cytoplasmic domain (e.g., Cvel_3066.t1). Representatives of this protein are found in the ciliate *Oxytricha*, but not in *Tetrahymena* and *Paramecium*; in stramenopiles, choanoflagellates and coccidians, but are absent from other apicomplexans such as *Cryptosporidium*, *Theileria*, *Babesia*, and *Plasmodium*.

## Appendix 6

## ApiAP2 proteins.

We examined the abundance of apicomplexan-specific AP2 (apiAP2) genes, transcription factors that play regulatory roles in key aspects of apicomplexan biology (*Campbell et al., 2010*; *Flueck et al., 2010*; *Radke et al., 2013*; *Kafsack et al., 2014*; *Sinha et al., 2014*). We scanned the protein-coding gene sets of the 26 species using the apicomplexan-specific apiAP2 HMM, which was constructed with the AP2 domain sequences from apicomplexan species. We found that apiAP2 genes were abundant in both chromerids and in all apicomplexans. ApiAP2 genes were moderately abundant in the two dinoflagellates and rare or absent in ciliates and stramenopiles, respectively (*Figure 3—figure supplement 1D*). There were very few apiAP2 genes that were shared between apicomplexans and non-apicomplexan species; most were shared between closely related species, that is, from the same clade (*Figure 3—figure supplement 1B*). These lineage-specific apiAP2 genes in the present-day species could have arisen from de novo gene birth or modification of the full-length sequences of existing genes beyond recognition. In the former case, the proto-apicomplexan ancestor had a small set of apiAP2 genes. In the latter case, the common ancestor already had a large set of apiAP2 genes, which continued to change, giving the appearance of 'new' clade-specific genes. The latter case, the turnover scenario, is more parsimonious because, according to the de novo gene birth scenario, apiAP2 genes must have expanded independently in every descending lineage from the proto-apicomplexan ancestor. In summary, our data support the notion that massive apiAP2 expansion occurred in the common ancestor before Apicomplexa and chromerids split, and the apiAP2s continued to change as the common ancestor split into chromerids and apicomplexans, which continued to radiate and adapt to their host niches and life cycle strategies.

We sought to determine if gene duplication and divergence was a significant mechanism for the expansion and the turnover of apiAP2 genes. The number of apiAP2 genes that have other homologous apiAP2 genes within the species based on OrthoMCL clustering, which are likely mediated by paralogous expansions, was high (93 out of 136) in chromerids (*Figure 3—figure supplement 1D*). In *Vitrella*, we identified one cluster of 50 homologous apiAP2 genes. This means that gene duplication played an important role in expanding apiAP2 gene repertoire in chromerids. The number was significantly less (5 out of 13) in dinoflagellates than in chromerids (*Figure 3—figure supplement 1D*). We suspect that gene duplication and diversifications drove expansion of apiAP2 genes significantly after the split of dinoflagellates. In apicomplexan species, evidence for recent duplications was sparse, as only 4 out of 409 apiAP2 genes had homologous copies in the same species. This does not necessarily mean that apiAP2 genes do not duplicate readily in apicomplexans, but rather that redundant copies of apiAP2 are quickly lost or diversified beyond recognition in part by selective pressure to reduce gene repertoires and genome sizes (*Katinka et al., 2001*) and due to higher rate of sequence divergence in parasites (*Hafner et al., 1994*).

Previous studies have shown that plant genomes contain a large repertoire of *AP2* genes, and that plant AP2 domains evolved from an endonuclease domain in a cyanobacteria (*Magnani et al., 2004*). According to our phylogenetic analysis, AP domains among bacteria are many and diverse, with both plant-like and apicomplexan-like AP2s (data not shown). We did not find significant homology with bacterial AP2 genes at the full gene length level. It is not clear if the originally transferred AP2 gene has evolved beyond recognition or if only the domain has been transferred to these eukaryotes. The exact genetic events that led to acquisition of AP2s in apicomplexans are not clear. However, what is the most probable scenario is that AP2 domains in alveolates came from bacteria and have expanded in myzozoans, independent of those in plants. Both functional studies and more taxon sampling would be required for elucidating how AP2s in alveolates were acquired in the first place.

