## [Decision Letter]

Thank you for sending your work entitled “Chromerid genomes reveal the evolutionary path from photosynthetic algae to obligate intracellular parasites” for consideration at *eLife*. Your article has been favorably evaluated by Detlef Weigel (Senior editor), Magnus Nordborg (Reviewing editor), and three reviewers, one of whom, Sophien Kamoun, has agreed to reveal his identity.

The Reviewing editor and the reviewers discussed their comments before we reached this decision, and the Reviewing editor has assembled the following comments to help you prepare a revised submission. The original comments are also included below, as they contain various specific comments you would be advised to take into account. However, the main points to be addressed in your response are the ones contained in the following summary.

Summary statement:

The importance of the new Chromerid genomes is the light they shed on the evolution of the *Apicomplexa*. The paper presents these genomes, and illustrates the kinds of insights that might be gained. The RNA-seq analysis, in particular, was innovative, and interesting. Unlike many genome papers, this one has a clear scientific question, and the paper should focus on it. The genomes themselves are essentially materials and methods, and should be described accordingly (no need for the kind of routine genome analyses used as filler in most genome papers). Essentially make it a rather short paper, perhaps with a paragraph each to describe genome assembly and annotation, AP2 homologs and extracellular and motility genes, slightly expand the rest.

In this context, we were concerned about how much these genomes bring compared to already published out-groups. In particular, there seems to be quite a bit of overlap with Klinger et al., 2013, PLOS ONE, and Oberstaller et al., 2014, NAR (http://nar.oxfordjournals.org/content/early/2014/06/22/nar.gku500.full).

Please address these specifically, and demonstrate that there is genuinely novel insight from these additional genomes.

Given that *eLife* is a broad interest journal, please be more precise when discussing “pre-existing parasitism genes” and similar. You write as if it were surprising that genes implicated in parasitism were present in the ancestor, but where else could they have come from? Also, their ancestral function was obviously not parasitism. Clearly you understand this, as will many or most of your readers, but please avoid jargon nonetheless.

We were also surprised that no attempt was made to test for evidence of accelerated evolution of the genes implicated in parasitism.

*Reviewer #1*:

This study describes the generation and analysis of two Chromerid genomes. Due to their close evolutionary relationships to apicomplexan parasites, the provision of these genomes represent a valuable resource to understanding the early origins of the *Apicomplexa*. Perhaps the main take home message from this study, is that the genome of the ancestor of Chromerids and apicomplexans already encoded many genes implicated in parasitism. This struck this reviewer as similar to findings from the analysis of the genome from the choanoflagellate, *Monosiga brevicolis*, which suggested that the genome of the common ancestor of Metazoa and choanoflagellates contained many genes associated with multicellularity—perhaps the authors may wish to refer to this parallel in their own study (King et al. Nature 2008).

In general the analyses are well performed and I really liked Figure 3. Further, the authors present additional experimental data supporting some of their conclusions, which is above what is usual for a genome style paper. There were however some concerns over parts of the analyses and interpretations that this reviewer would like to see addressed.

There is a suggestion that parasites have lost many, otherwise, typically conserved genes, particularly those involved in metabolism and signalling. However, reading through the Methods, if the annotations used to perform these comparisons were consistent. For example in Methods, it appears genes were annotated in the Chromerid genomes through the use of BLASTP, while for *Plasmodium* and *Toxoplasma*, there appears to be a reliance on ToxoDB and PlasmoDB which are more highly curated. Consequently, there is a suspicion that the larger number of conserved functions in Chromerids may be partly biased by annotation quality.

In the last paragraph of the subsection headed “Ancestral gene content of free-living and parasitic species”, there is a suggestion that apicomplexans have undergone massive gene loss due to a switch to a parasitic lifestyle. This is based on the presence and absence of large numbers of orthogroups. However an alternative explanation is that instead of losing many of these so called ‘core’ genes, they have simply undergone significant sequence divergence in the apicomplexan lineage, which might not be so surprising given the considerable changes in host tropism, life cycle, morphology etc. (in the first paragraph of the aforementioned subsection). The authors may be correct in their interpretation, however, this reviewer found the description of orthogroupings particularly confusing. For example, can orthogroupings contain singletons? If not, where do these appear in these analyses—can they be included in Figure 2?

Figure 3 shows a very nice depiction of the timing of emergence of important parasite gene families. However, some of the discussion in this section seems at odds with the earlier discussion on massive loss of orthogroups. Again this may be due to confusion over what constitutes an orthogrouping. That said, the contrast between these sections needs to be addressed (orthogroupings highlight gene loss, here gene gain is important).

The relevance of 100 horizontal gene transfers to the study wasn't made clear (second paragraph of the subsection headed “Genome assembly and annotation”).

“Apicomplexan genomes have significantly fewer genes than either chromerids […] consistent with the notion that reduction of cellular capabilities by gene losses occur as parasites evolve from free-living species” (in the first paragraph of the subsection headed “Progressive, lineage-specific losses during apicomplexan evolution”). This is a little misleading as parasites also acquire new genes to support e.g. host invasion, I would suggest rewording this.

What is known about the role of EC protein orthologs in Chromerids? Do they have secretory tags or have they been adapted for secretion by apicomplexans?

*Reviewer #2*:

The paper reports an interesting pair of chromerid genomes of free-living non-parasitic photosynthetic algae that are allied to apicomplexan parasites. These genomes add to our understanding of evolution of parasitism in Apicomplexa, a group that includes thousands of parasitic species. The main finding is that *Chromera* and *Vitrella* have dramatically larger gene repertoires and the authors conclude that parasitic species experienced major gene losses (∼3,800 orthologous groups). They also found that most core parasite genes were already present in the protoapicomplexan ancestor.

This is an interesting paper that I enjoyed reading. Gene loss in parasites is a well-established concept but the paper is comprehensive and should be useful to evolutionary biologists and parasitologists alike.

*Reviewer #3*:

I am not a specialist in this field. As an outsider, some of my concerns may be slightly naïve or misplaced, but I suppose I would argue that if they are concerns of mine, they may well also be concerns for the general reader of *eLife*.

The new data reported in this manuscript are two algal genomes (*Chromera velia* and *Vitrella brassicaformis*), and transcriptomes from these organisms in different conditions. These are important organisms as they are sister to the *Apicomplexa*, a group containing many devastating animal parasites. The central goal of the paper, then, is to infer what kind of changes (gene loss, gene gain, genome evolution) have taken place in the transition from free-living to parasitic lifestyles.

What I am struck by after several readings is how much of the analyses focus on apicomplexan organisms. I suppose this is natural, given the importance of these organisms, but it made me wonder how much of this paper was a re-analysis of data which are already published, since no apicomplexan data were actually generated for this work. In particular, since the genomes of *Symbiodinium* and *Perkinsus* are available and are really do not appear to be too much for divergent from *Apicomplexa* compared with *Vitrella* and *Chromera* (at least as drawn in Figure 1 of this paper), I wondered what insight was new here and what was a rehashing of old data (with, of course, the addition of two new species). As such, I looked into a couple of the specific points mentioned in the Abstract, the evolution of the endomembrane trafficking system and the AP2 proteins, and how the results described in this paper relate to other published work.

The similarities are pretty substantial. Taking the first example of the trafficking system, Figure 2—figure supplement 3 is very, very similar to Figure 4 from Klinger et al. 2013 PloS One 8: e76278. Not only are the figures similar, but so are the conclusions. I suppose this is not surprising, given that the senior author on the PLoS One paper is also a co-author on this manuscript, but it does concern me that much of the other results of this paper are not really that new or surprising. It argues against the claims of novelty reported in this paper, i.e., “Analysis of *Chromera* and *Vitrella* genomes has enabled the first genomic insights into how apicomplexan parasites have evolved from free-living ancestors, clearly delineating genes gained and lost at various stages of the evolution.” I would say that the data reported here seem to have “enabled the finest-scale genomic insights” of what genes have been lost during the evolution of parasitism, but seems to have not “enabled the first genomic insights”.

This concern is also relevant in the discussion of AP2 evolution. For example, in Balahi et al. 2005 Nucleic Acids Research 33: 3994, the authors state that “Detailed analysis of the apicomplexan genomes of *Plasmodium falciparum* and *Cryptosporidium parvum* showed that they entirely lacked conserved DNA binding domains of specific TFs found across the eukaryotic crown group, such as the homeo, bZip, bHLH and Fkh domains,” and “Thus, starting from a core set of at least nine proteins inherited from the ancestral form, the ApiAP2 family appears to have proliferated further through independent duplications as the different apicomplexan lineages emerged”. This seems to be very similar to what is stated in the present manuscript, e.g.: “Gains of apiAP2 genes, together with general reduction in other DNA binding proteins (Figure 2—figure supplement 1), appeared to have led to the dominant role of apiAP2 genes in present-day apicomplexans.” This conclusion seems to have been easily drawn with data available before the manuscript under consideration here (and the Balahi NAR paper was not cited).

---

## [Author Response]

*Summary statement*:

*The importance of the new Chromerid genomes is the light they shed on the evolution of the* Apicomplexa*. The paper presents these genomes, and illustrates the kinds of insights that might be gained. The RNA-seq analysis, in particular, was innovative, and interesting. Unlike many genome papers, this one has a clear scientific question, and the paper should focus on it. The genomes themselves are essentially materials and methods, and should be described accordingly (no need for the kind of routine genome analyses used as filler in most genome papers). Essentially make it a rather short paper, perhaps with a paragraph each to describe genome assembly and annotation, AP2 homologs and extracellular and motility genes, slightly expand the rest*.

We have reorganized this paper to be compact and focused with a single narrative. We have reduced the “routine” genome section to a single paragraph. We have removed 9 out of 15 sections of supplementary discussions (now called appendices) that contribute little to the central narrative. We drastically reorganized and streamlined the section titled “Emergent features of apicomplexans”, which originally contained paragraphs of the AP2 homologs and extracellular and motility genes, for which only findings that support our narratives are highlighted now. We have elaborated the section on the RNA-seq analysis further. As a result, our manuscript describes a study that focuses on a scientific question using the chromerid genomes and diverse transcriptomes as study materials.

*In this context, we were concerned about how much these genomes bring compared to already published out-groups. In particular, there seems to be quite a bit of overlap with Klinger et al., 2013, PLOS ONE, and Oberstaller et al., 2014, NAR (**http://nar.oxfordjournals.org/content/early/2014/06/22/nar.gku500.full**)*.

*Please address these specifically, and demonstrate that there is genuinely novel insight from these additional genomes*.

We appreciate the concerns raised by the Reviewing editor and Reviewer #3. The chromerid genomes and transcriptomes were indispensible for the new insights on the evolution of apicomplexan parasites. Dinoflagellates, other potential outgroup to apicomplexans, have diverged much further away from apicomplexans than chromerids have, and have undergone drastic, unique genomic and cellular changes (Janouškovec, Horák et al. 2010, Oborník, Modrý et al. 2012, Obornik and Lukes 2013). The available *Perkinsus marinus* genome sequence is highly fragmented, and there is no peer-reviewed paper describing the genome, and *Symbiodinium* genome sequence covers only 616Mb of the estimated 1.5Gb of the whole genome, according to the authors’ own estimate (Shoguchi, Shinzato et al. 2013). The low quality of their genomes hampers their utility as primary outgroups.

Without the chromerid genomes, i.e. with ciliate and dinoflagellate genomes only, our conclusions would have been vastly different. First, we tested one of our main claims that very few genes were gained without the chromerid genomes or the dinoflagellates genomes. The apicomplexan ancestor comprised 4859 orthogroups, of which only 81 were gained during Stage II. Without the chromerid genomes, 785 out of the 4859 orthogroups would have been inferred as gained after Stage I. Because of the chromerid genomes, we were able to conclude that most of the genes present in the apicomplexan ancestor were present in the free-living apicomplexan ancestor. In contrast, without the dinoflagellate genomes, only 207 orthogroups (compared to 785 without chromerids) would have been inferred as gained. Second, we identified 260 *Toxoplasma gondii* genes that are directly or indirectly implicated in parasitic processes (the categories listed in Figure 3) and share homology with free-living species. 27 were shared with the chromerid genome but not in the other free-living species. Examples of such genes include key components of gliding motility (GAP40 (Frenal, Polonais et al. 2010) and GAPM2 (Bullen, Tonkin et al. 2009)) and the inner membrane complex (ISP). In contrast, there were only two genes shared with the dinoflagellate genomes but not with the chromerid genome. While the dinoflagellate genomes reinforced our conclusion, it would have led us to the conclusion we reached in this study.

Also, we have generated comprehensive gene expression resources across 36 unique conditions based on RNA-seq, providing functional genomic information for the outgroup to the apicomplexans. In fact, the cross-species gene co-expression network analysis was indispensible for conclusions drawn in the study.

As for overlap with the Klinger et al. 2013 paper, although our figure (Figure 2—figure supplement 3) appear reminiscent of the previously published figure, the actual information included are drastically more, and the insights drawn are quite different from the previous study. Specifically, the 2013 paper analyzed only the MTCs, one of the three-endomembrane system gene sets analyzed in this manuscript. Our analysis also expanded the taxon sampling as compared with the 2013 paper and previous evolutionary analyses of membrane-trafficking machinery in apicomplexans. This enabled several new insights. Firstly, with the chromerid genomes in our analyses, we have concluded that some of the components missing in apicomplexans and other alveolates are not due to ancestral loss but due to multiple, independent losses in the various alveolate lineages. This is the case for Vps37 in the ESCRTs and several COG subunits in the MTCs, among others. Secondly, we now clarify that the proto-apicomplexan ancestor had a nearly complete endomembrane repertoire. This pinpoints that the progressive losses of endomembrane components (e.g. ESCRT 1 and 2, AP3, and COG) began after the chromerid apicomplexa split, concurrent with the transition to parasitism. By showing the presence of this gene set in the proto-apicomplexan, we better understand the dynamics of gene loss as occurring gradually and differentially by apicomplexan lineages in adaptation to specific niches. Thirdly, in direct contrast to the results from the [75] PLoS Biology paper first reporting the AP5 complex, we find AP5 to be widespread amongst apicomplexan lineages, including in the malaria parasite. Nonetheless, to address the fact that there were concerns about the novelty of the endomembrane analyses based on the three endomembrane protein sets analyzed, we have now included in the revised manuscript data from two additional endomembrane component sets implicated in biogenesis of the invasion organelles (Tomavo, Slomianny et al. 2013). The manuscript has now been revised to highlight all of these points.

With reference to the Oberstaller et al.*,* 2014 and [11] studies, our analyses based on the chromerid genome offers several insights not observed by past studies. It was previously thought that expansion of apiAP2 have occurred by independent duplication in different apicomplexan lineages, as Reviewer #3 has pointed out from [11]: *“*…the ApiAP2 family appears to have proliferated further through independent duplications as the different apicomplexan lineages emerged”. While our analyses confirmed this observation, our inclusion of chromerids addressed a very important, different question: did the free-living proto-apicomplexan ancestor heavily utilized apiAP2 proteins like present-day apicomplexans? Our findings support the notion that apiAP2 have drastically expanded in the proto-apicomplexans (Figure 3—figure supplement 1; Appendix 6). We observed a small repertoire of apiAP2 proteins in *Perkinsus* compared to that in chromerids, again underscoring the indispensability of the chromerid genomes for our conclusion. Also, the proto-apicomplexan ancestor genome encoded many proteins with canonical DNA Binding Domains (DBDs), suggesting that apiAP2 proteins were major, but not dominant, regulators of gene expression, and that the dominance of apiAP2 came after subsequent loss of proteins with DBDs during and after transition to apicomplexans (Figure 3—figure supplement 1). Our study for the first time pinpointed the sequence of evolutionary events because of the chromerid genomes. Importantly, we found evidence for gene duplication driving evolution of apiAP2, which has been postulated by Balaji et al. but not observed before, addressing how a large repertoire of such critical drivers of parasitism could have arisen (Appendix 6).

*Given that* eLife *is a broad interest journal, please be more precise when discussing “pre-existing parasitism genes” and similar. You write as if it were surprising that genes implicated in parasitism were present in the ancestor, but where else could they have come from? Also, their ancestral function was obviously not parasitism. Clearly you understand this, as will many or most of your readers, but please avoid jargon nonetheless*.

We thank and agree with the Reviewing editor for this valuable comment. We did not articulate the concept well in the manuscript which have led to the confusion on why would a non-parasite have ‘parasitism’ genes? We have modified the text, avoiding using jargons without definitive meanings, to convey the intended concept behind “pre-existing parasitism genes”: the group of genes (e.g. flagellar genes, some EC proteins) that were repurposed (or neo-functionalized) from their normal cellular roles in free-living organisms to molecular processes that are essential for life cycle of obligate parasites, including intracellular invasion.

In the manuscript, we also clarify a related concept that needs extra clarifications. We have concluded that many of the components that are now effectively exploited by apicomplexans as highly successful parasites were gained in this lineage prior to their commitment to parasitism. The section describing putative functions of genes based on RNA-seq analysis provide some clues to why they were present amongst the free-living, photosynthetic ancestors, represented by the chromerids.

*We were also surprised that no attempt was made to test for evidence of accelerated evolution of the genes implicated in parasitism*.

It is an interesting, important question whether adaptation to parasitism would correlate with accelerated evolution of genes involved in parasitic mechanisms (Woolfit and Bromham 2003, Castillo-Davis, Bedford et al. 2004, Bromham, Cowman et al. 2013, Duchene and Bromham 2013). However, it is impractical to test for this in apicomplexans for two reasons. First, a measure of accelerated evolution requires establishment of a basal rate of evolution within the lineages being examined. However, apicomplexans, and related lineages dinoflagellates and ciliates, are notorious for their overall high evolutionary rates. This is evident by the long branches in all of these groups in Figure 1 and is consistent with most published phylogenies. The reasons for this are unknown, but this does not correlate with parasitism alone. Second, while accelerated evolution of molecules can be associated with specialization, in this case for parasitism, it is also associated with loss of function and gradual accumulation of mutations. In apicomplexans we see multiple independent cases of pathway loss; indeed this is one of the important observations of this study. Thus, these two independent drivers of accelerated evolution in apicomplexans would further confound efforts to measure relative evolutionary rate associated with adaption to parasitism.

*Reviewer #1*:

*This study describes the generation and analysis of two Chromerid genomes. Due to their close evolutionary relationships to apicomplexan parasites, the provision of these genomes represent a valuable resource to understanding the early origins of the* Apicomplexa*. Perhaps the main take home message from this study, is that the genome of the ancestor of Chromerids and apicomplexans already encoded many genes implicated in parasitism. This struck this reviewer as similar to findings from the analysis of the genome from the choanoflagellate,* Monosiga brevicolis*, which suggested that the genome of the common ancestor of Metazoa and choanoflagellates contained many genes associated with multicellularity—perhaps the authors may wish to refer to this parallel in their own study (King et al. Nature 2008)*.

We thank the reviewer for this additional insight drawn from a previous study. Indeed, just as genome of choanoflagellates shed light in pre-adaptations to multicellularity, the genome of chromerids reveal how machinery present in free-living algae may have been co-opted and specialized to facilitate a parasitic lifestyle. However, while King et al. deals with evolution of multi-cellularity in metazoans from unicellular choanoflagellates (increased organismal complexity), our study deals with evolution of parasites from free-living species (mostly loss of complexity but increase in host-adaptive specialization). While analogy could be drawn, the comparison may create unnecessary, unintended confusions to a broad audience.

*In general the analyses are well performed and I really liked*
Figure 3*. Further, the authors present additional experimental data supporting some of their conclusions, which is above what is usual for a genome style paper. There were however some concerns over parts of the analyses and interpretations that this reviewer would like to see addressed*.

We are glad to note the complimentary words from the Reviewer #1.

*There is a suggestion that parasites have lost many, otherwise, typically conserved genes, particularly those involved in metabolism and signalling. However, reading through the Methods, if the annotations used to perform these comparisons were consistent. For example in Methods, it appears genes were annotated in the Chromerid genomes through the use of BLASTP, while for* Plasmodium *and* Toxoplasma*, there appears to be a reliance on ToxoDB and PlasmoDB which are more highly curated. Consequently, there is a suspicion that the larger number of conserved functions in Chromerids may be partly biased by annotation quality*.

The chance of annotation bias across species is minimal because for most of the analyses in the study, we applied the same computational prediction method with the same threshold to the 27 genomes. For example, identification of functional protein domains (Pfam) was conducted by HMMER applied uniformly against the 27 genomes, and KEGG annotation of metabolic enzymes was based on BLASTP against all of the 27 genomes. For the analyses of evolutionary ages of *Plasmodium* and *Toxoplasma* genes, we have curated the genes in *Toxoplasma* mostly by computational prediction method (again which were uniformly applied to all sequences in all species; for example, HMMER search of the RAP genes, apiAP2 genes, other DNA-binding domain, other RNA-binding domains, alveolins, actins and myosins).

*In the last paragraph of the subsection headed “Ancestral gene content of free-living and parasitic species”, there is a suggestion that apicomplexans have undergone massive gene loss due to a switch to a parasitic lifestyle. This is based on the presence and absence of large numbers of orthogroups. However an alternative explanation is that instead of losing many of these so called ‘core’ genes, they have simply undergone significant sequence divergence in the apicomplexan lineage, which might not be so surprising given the considerable changes in host tropism, life cycle, morphology etc. (in the first paragraph of the aforementioned subsection). The authors may be correct in their interpretation, however, this reviewer found the description of orthogroupings particularly confusing. For example, can orthogroupings contain singletons? If not, where do these appear in these analyses—can they be included in*
Figure 2?

We agree that we must consider the current conclusion that there were massive losses of genes against the alternative explanation that sequences have diverged to a point that no homology would be detected. The drastic net loss of genes in apicomplexans compared to chromerids (Figure 2) would support the explanation of gene loss. Also, very specific set of genes were lost or conserved; for example, while enzymes of the TCA cycle were conserved, most of the photosynthesis-related components were lost; we argue that a large set of genes were lost because they are no longer necessary, rather than that they have evolved into genes with new functions. Although we cannot completely rule out cases consistent with the alternative explanations, we find that for most of the cases our explanation is more parsimonious and consistent with the findings of this study.

Orthogroups refers to a group of homologous sequences across all genes and across all species. They can be one sequence (singleton), or a group of paralogs and/or orthologs depending on the evolutionary event. While the term orthogroup does not describe the precise evolution event, unable to distinguish orthology from paralogy, it is extremely practical and powerful because it allows us to collectively describe a singleton, putative orthologs, and paralogs. The term orthogroup is being increasingly used in the literature (Amborella Genome 2013, Lovell, Wirthlin et al. 2014, Dey, Jaimovich et al. 2015). Gains and losses of the orthogroups are shown in Figure 2; the number of orthogroup at each node was not displayed because the figure would then be too busy and complex.

Figure 3
*shows a very nice depiction of the timing of emergence of important parasite gene families. However, some of the discussion in this section seems at odds with the earlier discussion on massive loss of orthogroups. Again this may be due to confusion over what constitutes an orthogrouping. That said, the contrast between these sections needs to be addressed (orthogroupings highlight gene loss, here gene gain is important)*.

We thank the reviewer for bringing up this issue. We have clarified the definition of ‘orthogroup’ above and in the manuscript text (in the second paragraph of the subsection headed “Ancestral gene content of free-living and parasitic species”). In Figure 2, we examine all of the orthogroups that were present in at least one of the 27 species, and see their ‘gains’ and ‘losses’ of orthogroups during apicomplexan evolution. In Figure 3, we identify evolutionary ages, i.e. the time of gain, of genes in the present-day apicomplexans. Essentially, Figure 3 represents a subset of information presented in Figure 2. We found this to be necessary as we intend to give a focused narrative. We have revised the manuscript to better articulate this.

*The relevance of 100 horizontal gene transfers to the study wasn't made clear (second paragraph of the subsection headed “Genome assembly and annotation”)*.

Since both *Chromera velia* and *Vitrella brassicaformis* in their natural marine habitat live in close proximity to bacteria and other organisms (e.g. coral metaorganism), we have initially wanted to examine the extent of laterally transferred genes in both genomes. We reported over 100 independent lateral gene transfers in each species, contributing to the divergent gene repertoires between the two chromerids. Based on editorial suggestions, we have now removed this section out of the manuscript as they do not contribute substantially to the main conclusion of this study.

*“Apicomplexan genomes have significantly fewer genes than either chromerids […] consistent with the notion that reduction of cellular capabilities by gene losses occur as parasites evolve from free-living species” (in the first paragraph of the subsection headed “Progressive, lineage-specific losses during apicomplexan evolution”). This is a little misleading as parasites also acquire new genes to support e.g. host invasion, I would suggest rewording this*.

To clarify that this is not the only major event, we have revised it as follows: “one of the major events during apicomplexan evolution is progressive, continued losses of components important for free-living organisms”.

*What is known about the role of EC protein orthologs in Chromerids? Do they have secretory tags or have they been adapted for secretion by apicomplexans*?

As shown in Figure 3—figure supplement 4 and described in Appendix 5, chromerid EC proteins do contain canonical secretory tags and often show remarkable similarity in the protein domain combination architectures to apicomplexan EC proteins, although gene-level orthology are rare. Atypical secretory tags (e.g. PEXEL motifs in *Plasmodium* EC proteins) are not found in Chromerid EC proteins. Those EC proteins with atypical apicomplexan export signatures (e.g. VAR, RIF, PHIST genes) are not shared with chromerids and they were gained at a later stage (after stage III, as depicted in Figure 2) of the evolutionary innovations in apicomplexan parasites.

Reviewer #3:

*I am not a specialist in this field. As an outsider, some of my concerns may be slightly naïve or misplaced, but I suppose I would argue that if they are concerns of mine, they may well also be concerns for the general reader of* eLife*.*

*The new data reported in this manuscript are two algal genomes (*Chromera velia *and* Vitrella brassicaformis*), and transcriptomes from these organisms in different conditions. These are important organisms as they are sister to the* Apicomplexa*, a group containing many devastating animal parasites. The central goal of the paper, then, is to infer what kind of changes (gene loss, gene gain, genome evolution) have taken place in the transition from free-living to parasitic lifestyles*.

*What I am struck by after several readings is how much of the analyses focus on apicomplexan organisms. I suppose this is natural, given the importance of these organisms, but it made me wonder how much of this paper was a re-analysis of data which are already published, since no apicomplexan data were actually generated for this work. In particular, since the genomes of* Symbiodinium *and* Perkinsus *are available and are really do not appear to be too much for divergent from* Apicomplexa *compared with* Vitrella *and* Chromera *(at least as drawn in*
Figure 1
*of this paper), I wondered what insight was new here and what was a rehashing of old data (with, of course, the addition of two new species). As such, I looked into a couple of the specific points mentioned in the Abstract, the evolution of the endomembrane trafficking system and the AP2 proteins, and how the results described in this paper relate to other published work*.

*The similarities are pretty substantial. Taking the first example of the trafficking system,*
Figure 2—figure supplement 3
*is very, very similar to*
Figure 4
*from Klinger et al. 2013 PloS One 8: e76278. Not only are the figures similar, but so are the conclusions. I suppose this is not surprising, given that the senior author on the PLoS One paper is also a co-author on this manuscript, but it does concern me that much of the other results of this paper are not really that new or surprising. It argues against the claims of novelty reported in this paper, i.e., “Analysis of* Chromera *and* Vitrella *genomes has enabled the first genomic insights into how apicomplexan parasites have evolved from free-living ancestors, clearly delineating genes gained and lost at various stages of the evolution.” I would say that the data reported here seem to have “enabled the finest-scale genomic insights” of what genes have been lost during the evolution of parasitism, but seems to have not “enabled the first genomic insights”*.

*This concern is also relevant in the discussion of AP2 evolution. For example, in*
*Balahi et al. 2005*
*Nucleic Acids Research 33: 3994, the authors state that “Detailed analysis of the apicomplexan genomes of Plasmodium falciparum and* Cryptosporidium parvum *showed that they entirely lacked conserved DNA binding domains of specific TFs found across the eukaryotic crown group, such as the homeo, bZip, bHLH and Fkh domains,” and “Thus, starting from a core set of at least nine proteins inherited from the ancestral form, the ApiAP2 family appears to have proliferated further through independent duplications as the different apicomplexan lineages emerged”. This seems to be very similar to what is stated in the present manuscript, e.g.: “Gains of apiAP2 genes, together with general reduction in other DNA binding proteins (*Figure 2—figure supplement 1*), appeared to have led to the dominant role of apiAP2 genes in present-day apicomplexans.” This conclusion seems to have been easily drawn with data available before the manuscript under consideration here (and the Balahi NAR paper was not cited)*.

In our opinion, the manuscript, in its revised form, provides unprecedented insights into how obligate intracellular parasites of terrestrial animals and human have evolved from free-living marine algae. Our detailed genomic, transcriptomic and cell-biological analyses of *Chromera velia* and *Vitrella brassicaformis* reveal that free-living algal ancestors possessed the molecular toolkits that when rewired and repurposed gradually giving rise to disease-causing apicomplexan parasites, including those that cause malaria, toxoplasmosis and cryptosporidiosis. We provide a comprehensive roadmap of evolutionary changes characterized by gene loss and cellular innovations associated with parasitism during the genesis and radiation of *Apicomplexa*. Hence, we firmly believe that this study is of significant interest to a wide audience in *eLife*. We have addressed all of the concerns raised by Reviewer #3 as a part of response to the Summary statement of the editors above.